# InfoPrompt: Information-Theoretic Soft Prompt Tuning for Natural Language Understanding

**Junda Wu**[1*]    **Tong Yu**[2*]    **Rui Wang**[3]    **Zhao Song**[2]    **Ruiyi Zhang**[2]
**Handong Zhao**[2]    **Chaochao Lu**[4†]    **Shuai Li**[5‡]    **Ricardo Henao**[3,6]
[1]University of California, San Diego    [2]Adobe Research    [3]Duke University
[4]University of Cambridge    [5]Shanghai Jiao Tong University    [6]KAUST
juw069@ucsd.edu
{tyu,zsong,ruizhang,hazhao}@adobe.com
{rw161,ricardo.henao}@duke.edu
cl641@cam.ac.uk    shuaili8@sjtu.edu.cn

## Abstract

Soft prompt tuning achieves superior performances across a wide range of few-shot tasks. However, the performances of prompt tuning can be highly sensitive to the initialization of the prompts. We have also empirically observed that conventional prompt tuning methods cannot encode and learn sufficient task-relevant information from prompt tokens. In this work, we develop an information-theoretic framework that formulates soft prompt tuning as maximizing the mutual information between prompts and other model parameters (or encoded representations). This novel view helps us to develop a more efficient, accurate and robust soft prompt tuning method, InfoPrompt. With this framework, we develop two novel mutual information based loss functions, to (i) explore proper prompt initialization for the downstream tasks and learn sufficient task-relevant information from prompt tokens and (ii) encourage the output representation from the pretrained language model to be more aware of the task-relevant information captured in the learnt prompts. Extensive experiments validate that InfoPrompt can significantly accelerate the convergence of the prompt tuning and outperform traditional prompt tuning methods. Finally, we provide a formal theoretical result to show that a gradient descent type algorithm can be used to train our mutual information loss.

## 1 Introduction

Soft prompt tuning has shown great successes in a wide range of natural language processing tasks, especially in low-resource scenarios [60, 37, 66]. With a relatively small size of prompt parameters appended to the input of the context, the language model can be adapted to the downstream tasks with the large scale pretrained parameters frozen. Compared with conventional fine tuning methods, prompt tuning requires less memory and computational resources to update these significantly smaller sized prompt parameters. In addition, in low-shot learning scenarios, prompt tuning can prevent the language model from overfitting on the training data, thus maintaining the generalization ability of pretrained language models.

However, recent works reveal that it is non-trivial to find a proper initialization of the prompt tokens. Several works have investigated the effect of prompt initialization on the prompt tuning performances

---

[*]These authors contributed equally to this work.

[†]The work was done previously when this author was at the University of Cambridge. This author is now at Shanghai AI Laboratory.

[‡]Corresponding author.

37th Conference on Neural Information Processing Systems (NeurIPS 2023).

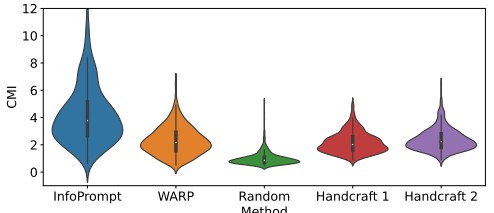 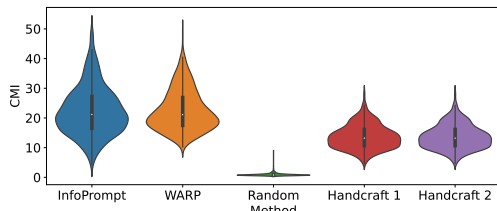

(a) MRPC. Prompt handcraft 1 is 'it is equivalent to' and prompt handcraft 2 is 'has the same meaning'.

(b) SST-2. Prompt handcraft 1 is 'this is positive' and prompt handcraft 2 is 'this is negative'.

Figure 1: Distributions of the CMI metrics of the prompts learned or handcrafted from different methods on MRPC and SST-2 [95]. By our InfoPrompt, the relevance between the prompt tokens and downstream tasks is the highest among all methods.

[91, 101] and showed that the performances of prompt tuning are highly sensitive to the prompt initialization. However, since the proper prompt initialization can vary to different downstream tasks and pretrained language models, it is hard to find very accurate knowledge to guide us to obtain the proper initialization [13].

In addition to the above limitations, , we also empirically observe that conventional prompt tuning methods cannot effectively learn sufficient task-relevant information from prompt tokens. Specifically, the prompts may fail to learn sufficient information that is relevant to the downstream tasks. To understand the relevance between the prompt tokens and downstream tasks, we calculate the conditional mutual information (CMI) between the prompt tokens and the latent representation from the language model conditioned on the input context. We follow [40] in determining the positions of prompt tokens inserted between the input sentences. Figure 1 shows the distribution of CMI of the prompts resulting from different methods. The randomly sampled prompts have the lowest CMI. The prompts learned by a soft prompt tuning method, WARP [40], can have relatively higher CMI than the handcrafted ones. By directly maximizing the CMI, our InfoPrompt (detailed in Section 3) facilitates learning of more informative prompts. Without the guidance of task-relevant information, randomly exploring the optimal prompts within the large continuous embedding space of the prompt tokens can be inefficient, *i.e.* a similar challenge is also discussed in [75]. Some related results [75, 37, 78] show that prompt tuning takes much larger numbers of epochs to converge than fine tuning. Comparatively, thanks of the guidance of CMI, our propose InfoPrompt allows prompt tuning to converge much faster. We also provide theoretical guarantees of the convergence of the proposed losses when training with gradient descent based algorithms.

Overall, we develop an information-theoretic framework that formulates soft prompt tuning as maximizing mutual information between prompts and other model parameters (or encoded representations), conditioned on the input context. With this framework, we develop InfoPrompt with two novel mutual information based loss functions. (i) To explore proper prompt initialization for the downstream tasks and learn prompts with sufficient task-relevant information, we optimize the *head loss* which maximizes the mutual information between the prompt and the task-specific classification head. By optimizing this head loss, the prompt can effectively learn task-relevant information from the downstream tasks, since the task-specific classification head usually contains the information from the downstream tasks. Besides, the mutual information loss can help to guide the learning of the prompt tokens in the early steps to tackle the initialization challenge, since the classification head parameters can learn the downstream task information more quickly. (ii) To further encourage the output representation from the pretrained language model (*i.e.*, encoded representation) to be more aware of the task-relevant information captured in the learnt prompt, we optimize the *representation loss* which maximizes the conditional mutual information between the prompt embeddings and the feature representations conditioned on the input context.

Our contributions are summarized as:

- We revisit the challenge of initialization with prompt tuning and show that existing prompt tuning methods fail to learn sufficient task-relevant information.

- We propose InfoPrompt, a framework to solve such challenges from a information-theoretic perspective. Specifically, we develop two novel loss functions that effectively find proper

prompt initialization and learn sufficient task-relevant information from down-stream tasks, without requiring any prior knowledge.

- Extensive experiments on multiple tasks and datasets validate that, InfoPrompt can significantly accelerate the convergence of the prompt tuning and outperform existing prompt tuning methods with higher accuracy.

- We provide a formal theoretical result to show that our proposed loss functions can be optimized using gradient descent based algorithm with convergence guarantees.

## 2 Preliminary

### 2.1 Prompt Tuning

Prompt tuning has shown great successes in a wide range tasks of NLP [60, 37, 66]. Let $\Phi$ denote the encoder of a pretrained language model, *e.g.*, the Roberta-Large [67]. Assume $X = \{x_1, x_2, \cdots, x_n\}$ is a length-$n$ text sequence of and $Y$ is its classification label. In prompt tuning, we add extra information $P$ for the encoder to condition on for its prediction of $Y$. $P = \{p_1, \ldots, p_{n_p}\}$ is a sequence of prompt embeddings and $n_p$ is the number of prompt tokens. $p_i \in \mathbb{R}^D$, $i = 1, \cdots, n_p$, is an embedding vector with dimension $D$ and $D$ is also the embedding dimension of the pretrained language model. We first embed each token of $X$ into its corresponding token embedding from the pretrained language model. $P$ is inserted into the resulting embedding sequence of $X$, and the resulting sequence is further encoded by the pretrained encoder $\Phi$ into the representation space of the pretrained language model. The template for inserting of prompt tokens is detailed in Section 4. Formally, we denote such a process by $Z = \Phi(P, X)$, with $Z$ being the output representation from the pretrained encoder. The model prediction for $X$ is made on top of $Z$ with a trainable classification head parameterized by $\theta$, denoted as $h_\theta$, whose output $h_\theta(Z)$ is the probability distribution over all possible classification labels. For classification, the prompts are trained via minimizing the following loss function,

$$\mathcal{L}_{\text{pred}} = \text{cross\_entropy}(h_\theta(Z), Y)$$

Parameters of the pretrained encoder $\Phi$ is frozen during training. Different from previous works (*e.g.*, [60]) where the prompts are directly learnt, the prompts in our approach are encoded from the input $X$. In this way, the resulting prompts can better capture task-relevant information from the training text $X$. We will elaborate on how the prompts are encoded from input $X$ in Section 4.

### 2.2 Mutual Information

Mutual information (MI) is a metric in information theory [79, 19], which quantifies the amount of information shared between two random variables. The mutual information between two random variables $A$ and $B$ is

$$\mathcal{I}(A; B) = \mathbb{E}_{p(a,b)} \left[ D_{KL} \left[ p(a|b) \| p(b) \right] \right].$$

Inspired by [89], we use MI as the criterion for comparing prompts and other model parameters. MI has also been applied to measure the similarity between the masked token and the corresponding context in the pretraining of Multilingual Masked Language Modeling [15], the relevance between documents and sentences in document summarization [72] and the source and target sentences in Neural Machine Translation (NMT) [107].

## 3 Our Method: InfoPrompt

As mentioned above, we want the learnt prompts to be task-relevant. To achieve this, we notice that the classification head is trained with both the data representation and the classification labels, thus should contain rich information of the learnt tasks. In encouraging the task-relevancy of the learnt prompts, we consider maximizing the mutual information between the prompt and the parameters of the classification head, denoted as $\theta$. By maximizing such mutual information, the learnt prompt will be more aligned with the training data with which the classification head is trained, thus captures more task-relevant information from training. Further, in order for the pretrained language model to

properly leverage the task-relevant information in the prompt, we additionally maximize the mutual information between the prompt and the representation from the pretrained language model, so that the encoded representation can be aware of the task-relevant information captured by the prompt. In addition, we also provide theoretical guarantees of the convergence of those losses when training with gradient descent, demonstrating that our method can converge more easily than existing prompt tuning methods. Below, we denote the negative mutual information between the prompt and parameters of the classification head as the *head loss*. The negative mutual information between the prompt and representations from the pretrained language model is denoted as the *representation loss*.

## 3.1 The Head Loss

The head loss is the negative mutual information between the prompt $P$ and parameters $\theta$, *i.e.*, $-I(P; \theta|X)$. In maximizing $I(P; \theta|X)$, we follow [68] that approximates it with the following lower bound,

$$\mathcal{I}(P; \theta|X) \geq C + \mathcal{L}_{NCE}(P, \theta, X),$$

where $C$ is a constant, $\mathcal{L}_{NCE}$ is a Noise Contrastive Estimation (NCE) of mutual information,

$$\mathcal{L}_{NCE} = \mathbb{E}\left[\log \frac{\exp(l(P, \theta, X))}{\sum_{k=1}^{K} \exp(l(P_k, \theta|X))}\right],$$

and $\{P_k\}_{k=1}^{K}$ are the negative prompt samples for contrastive learning. In practice, we randomly sample $K-1$ tokens from the context as the negative samples, *i.e.*, $\{P_k\}_{k=2}^{K}$, and the positive sample is $P_1 = P$.

We model the score function $l(P, \theta|X)$ as a standard bilinear function with the learnable matrix $W_1$

$$l(P, \theta|X) = P^\top W_1 \theta.$$

where $\theta$ and $P$ are encoded from $X$, and $W_1$ is a trainable matrix. Since the classification head is learnt on top of the output from the last layer of the pretrained language model, the learning of its parameters $\theta$ is easier than the learning of the prompt $P$ (the input layer of the pretrained language model). Therefore, $\theta$ may capture more task-relevant information than $P$ in the early stage of training. By maximizing the mutual information between $\theta$ and $P$, the task-relevant information can be transferred to $P$ in the initial training steps. In this way, $P$ can be more task-relevant especially in the early training stage. Experiments in Section 6 also show that our head loss, $\mathcal{I}(P; \theta|X)$, can facilitate the training of the initial training steps.

## 3.2 The Representation Loss

The representation loss, denoted as $-\mathcal{I}(P; Z|X)$, is defined as the negative of mutual information between the prompt $P$ and the encoded representation from the pretrained language model, *i.e.*, $Z = \Phi(P, X)$. Similar to the head loss, we approximate the representation loss with its lower bound,

$$\mathcal{I}(P; Z|X) \geq \log(N) + \mathcal{L}_{NCE}(P, Z|X),$$

and,

$$\mathcal{L}_{\text{NCE}} = \mathbb{E}\left[\log \frac{\exp(l(P, Z|X))}{\sum_{k=1}^{K} \exp(l(P, Z_k|X))}\right],$$

$\{Z_k\}_{k=1}^{K}$ are the negative samples. Here, we overload the notations of InfoNCE loss $\mathcal{L}_{NCE}$ and score function $l$ for conciseness. Let $W_2$ be a trainable matrix, the function $l$ for the representation loss is defined by,

$$l(P, Z|X) = P^\top W_2 Z.$$

We use variational inference methods [46] to recover the latent distribution of $Z$. Specifically, we assume that the latent distribution is $N(\mu, \sigma)$, where $N(\mu, \sigma)$ is the normal distribution with mean $\mu$ and diagonal covariance matrix $\sigma$. We model $\mu$ and $\sigma$ via,

$$\mu = f_\mu(Z), \sigma = f_\sigma(Z).$$

$f_\mu$ and $f_\mu$ are trainable fully connected layers. Since the negative samples of $Z$, *i.e.*, $\{Z_k\}_{k=1}^K$, should not be paired with $P$, we assume the $\{Z_k\}_{k=1}^K$ are drawn from $N(\mu', \sigma')$, *s.t.*,

$$\mu' = f_\mu(Z'), \sigma' = f_\sigma(Z').$$

In contrast to $Z = \Phi(P, X)$, we have $Z' = \Phi(X)$ where $\{Z_k\}_{k=1}^K$ are not paired with $P$. By maximizing the representation loss $I(P; Z|X)$, we encourage the encoded representation $Z$ to be more aware of the prompt $P$, so that the task-relevant information in $P$ can be properly encoded by the pretrained language model in producing $Z$.

### 3.3 Overall Objective

We minimize the following objective in prompt tuning:

$$\mathcal{L} = \mathcal{L}_{\text{pred}} - \beta \cdot \mathcal{I}(P; Z|X) - \gamma \cdot \mathcal{I}(P; \theta|X). \tag{1}$$

We denote $\mathcal{L}_{\text{pred}}$ as the task loss. $\beta$ and $\gamma$ are balancing parameters for the proposed representation loss and head loss, respectively. We denote our approach as *InfoPrompt*. More details about the implementation and configurations are provided in Section 4.2.

### 3.4 Theoretical Guarantees

We state our main theoretical result as follows. Due to the space limit, we delay the proof into Appendix.

**Theorem 3.1.** *Given the Loss function $\mathcal{L}$ (Eq. (1)) and conditions specified in Appendix C.1 and D, using gradient descent type of greedy algorithm, we can find the optimal solution of that loss function.*

We provide theoretical guarantees of the convergence of those losses trained by conventional gradient descent type algorithms. In Section 4, we empirically observe that our method converges more easily than traditional soft prompt tuning methods and requires fewer training epochs.

## 4 Experiments

### 4.1 Datasets

We conduct experiments with datasets of sequence classification from the GLUE benchmark [95], along with those of relation extraction tasks and NER tasks. We choose four sequence classification tasks from the GLUE benchmark: RTE (Recognizing Textual Entailment, [7]), MRPC (Microsoft Research Paraphrase Corpus, [28]), CoLA (Corpus of Linguistic Acceptability, [98]) and SST-2 (Sentence Sentiment Treebank, [81]). We choose these tasks because their datasets are of smaller sizes and prompt tuning is comparably more effective in low-resource settings [40, 62]. For the task of relation extraction, we evaluate our method on the ACE2005 corpus and the Semeval-2010 datasets [44]. We also use the ACE2005 corpus for the task of NER. Note that the entity spans for NER have been given ACE2005. Unlike the standard NER model that learns to predict the entity span and entity type simultaneously from the raw text sequence, our model only predicts the entity type based on the given entity span. We follow the same data splitting strategy for ACE2005 corpus as the previous work [103, 71]. For the Semeval-2010 tasks, we follow the official data partition [44].

### 4.2 Experiment Settings

We follow the resource constrained scenario in [40] that trains each task with only 64 or 256 samples. We experiment with $n_p = 1$ and $n_p = 4$ prompt tokens for each task. The prompt tokens are inserted into the template for each task. Similar to [40], we adopt the RoBERTa-large model as our pretrained encoder. We freeze the pretrained parameters and only train the parameters of the prompt head and prompt tokens. During training, we empirically set $\beta = 0.1$ and $\gamma = 0.05$. The number of negative samples is $K = 32$. The learning rate is $1e-3$ and the batch size is 8. For each task, we report the results after 30 epochs, averaged over 5 random seeds. To encode the prompt $P = [p_1, \cdots, p_{n_p}]$ from $X$, we first encode $X$ into $P' \in \mathbb{R}^D$ via $P' = \Phi(X)$. We denote the up-sampling and down-sampling

Table 1: Results on Sequence Classification.

| | CoLA | | RTE | | MRPC | | SST2 | | |
|---|---|---|---|---|---|---|---|---|---|
| | $n_p = 1$ | $n_p = 4$ | $n_p = 1$ | $n_p = 4$ | $n_p = 1$ | $n_p = 4$ | $n_p = 1$ | $n_p = 4$ | Average |
| Finetuning | 0.6131 | | 0.7798 | | 0.8873 | | 0.9427 | | 0.8057 |
| Adapter [45] | 0.5552 | | 0.5776 | | 0.6814 | | 0.9472 | | 0.6904 |
| WARP [40] | 0.5282 | 0.5911 | 0.6282 | 0.6426 | 0.8039 | 0.8186 | 0.9507 | 0.9587 | 0.7403 |
| IDPG [102] | 0.5556 | 0.5646 | 0.6282 | 0.6534 | 0.7941 | 0.8039 | 0.9587 | 0.9587 | 0.7396 |
| InfoPrompt | 0.5631 | 0.6018 | 0.6751 | 0.6968 | 0.8039 | 0.8137 | 0.9576 | 0.9599 | 0.7590 |
| $\gamma = 0$ | 0.5699 | 0.5853 | 0.6751 | 0.6787 | 0.7941 | 0.8137 | 0.9495 | 0.9587 | 0.7531 |
| $\beta = 0$ | 0.5546 | 0.5579 | 0.6065 | 0.6318 | 0.7892 | 0.7966 | 0.9472 | 0.9610 | 0.7306 |
| $\gamma = 0, \beta = 0$ | 0.5032 | 0.5732 | 0.6173 | 0.6029 | 0.7917 | 0.7672 | 0.9495 | 0.9564 | 0.7202 |

Table 2: Results on Relation Extraction and NER.

| | RE | | NER | | SemEval | | |
|---|---|---|---|---|---|---|---|
| | $n_p = 1$ | $n_p = 4$ | $n_p = 1$ | $n_p = 4$ | $n_p = 1$ | $n_p = 4$ | Average |
| Finetuning | 0.8119 | | 0.9054 | | 0.8506 | | 0.8560 |
| Adapter [45] | 0.5073 | | 0.8329 | | 0.6570 | | 0.6657 |
| WARP [40] | 0.6384 | 0.6596 | 0.8174 | 0.8607 | 0.6702 | 0.7284 | 0.7291 |
| IDPG [102] | 0.6079 | 0.6132 | 0.8360 | 0.8931 | 0.6408 | 0.6776 | 0.7114 |
| InfoPrompt | 0.6914 | 0.7616 | 0.8526 | 0.8962 | 0.7563 | 0.7917 | 0.7916 |
| $\gamma = 0$ | 0.6914 | 0.7285 | 0.8452 | 0.8635 | 0.7471 | 0.7865 | 0.7770 |
| $\beta = 0$ | 0.6967 | 0.7470 | 0.8351 | 0.8698 | 0.7449 | 0.7538 | 0.7746 |
| $\gamma = 0, \beta = 0$ | 0.5364 | 0.7285 | 0.8512 | 0.8661 | 0.7490 | 0.7799 | 0.7519 |

projections similar in [31]. For each $p_i \in \mathbb{R}^D$, we have $p_i = W_i^{\mathrm{up}} W_i^{\mathrm{down}} P'$, $W_i^{\mathrm{up}} \in \mathbb{R}^{D \times 64}$, $W_i^{\mathrm{down}} \in \mathbb{R}^{64 \times D}$.

For the tasks of sequence classification and relation extraction, we follow the template of [40] that contains a [mask] token. The representation $Z$ is obtained from the [mask] token from the last layer of the RoBERTa-Large encoder. For the task of NER, we have the [mask] token before the given entity span, with the rest being the same as for sequence classification.

## 4.3 Baselines and Ablations

As mentioned above, our method with Eq. (1) in denoted as InfoPrompt. In the experiments, we compare our method with the following baselines:

- Finetuning: We fine tune all the parameters from the pretrained encoder on each task. Finetuning is included as the upper bound for the model performance, since it is more computational expensive compared with only training the prompt parameters.

- Adapter [45]: Similar to prompt tuning, this is also a way of parameter-efficient training for pretrained language models. Specifically, instead of adding the prompt tokens in the input, we add adapters after the feed-forward module in each transformer layer.

- WARP [40]: Different from our approach, the prompt tokens of WARP are not generated from the input sequence. the prompt tokens are insert into the input sequence. During training, the pretrained encoder is frozen and only the prompt tokens are trainable.

- IDPG [102]: Similar to our approach, the prompt tokens are generated from the input sequence. The pretrained encoder is frozen and the prompt generator is trainable.

In evaluating the effectiveness of our proposed loss functions, we consider the following two ablations:

- $\gamma = 0$: We disable the head loss during training via $\gamma = 0$, while keeping $\beta = 0.05$.

- $\beta = 0$: We disable the representation loss during training via $\beta = 0$, while keeping $\gamma = 0.1$.

- $\beta = \gamma = 0$: We disable both the losses. The prompt parameters are trained with $L_{\mathrm{pred}}$.

Table 3: Few-shot results on Sequence Classification. We experiment with $N = 64$ and $N = 256$ samples for each task. The number of prompt is fixed to $n_p = 4$ for all soft prompt tuning methods.

| | CoLA | | RTE | | MRPC | | SST2 | | |
| | $N = 64$ | $N = 256$ | $N = 64$ | $N = 256$ | $N = 64$ | $N = 256$ | $N = 64$ | $N = 256$ | Average |
|---|---|---|---|---|---|---|---|---|---|
| Finetuning | 0.1746 | 0.4086 | 0.4801 | 0.6787 | 0.7107 | 0.7819 | 0.8027 | 0.8853 | 0.6153 |
| Adapter [45] | 0.0627 | 0.2486 | 0.5487 | 0.5668 | 0.5931 | 0.6250 | 0.4908 | 0.664 | 0.4750 |
| WARP [40] | 0.0749 | 0.0785 | 0.5596 | 0.5812 | 0.7083 | 0.7083 | 0.5872 | 0.7638 | 0.5077 |
| IDPG [102] | 0.0902 | 0.1513 | 0.5018 | 0.5523 | 0.6593 | 0.7010 | 0.5424 | 0.8188 | 0.5021 |
| InfoPrompt | 0.1567 | 0.1750 | 0.6137 | 0.6580 | 0.7059 | 0.7377 | 0.6697 | 0.7305 | 0.5559 |
| $\gamma = 0$ | 0.1479 | 0.1447 | 0.5776 | 0.6318 | 0.6936 | 0.7328 | 0.664 | 0.7294 | 0.5402 |
| $\beta = 0$ | 0.1372 | 0.1433 | 0.5812 | 0.5957 | 0.6838 | 0.7132 | 0.5631 | 0.656 | 0.5092 |
| $\gamma = 0, \beta = 0$ | 0.0919 | 0.1397 | 0.5668 | 0.5523 | 0.6985 | 0.7108 | 0.5505 | 0.6296 | 0.4925 |

Table 4: Few-shot results on Relation Extraction and NER. We experiment with $N = 64$ and $N = 256$ samples for each task. The number of prompt is fixed to $n_p = 4$ for all soft prompt tuning methods.

| | RE | | NER | | SemEval | | |
| | $N = 64$ | $N = 256$ | $N = 64$ | $N = 256$ | $N = 64$ | $N = 256$ | Average |
|---|---|---|---|---|---|---|---|
| Finetuning | 0.1285 | 0.4013 | 0.3033 | 0.4358 | 0.2223 | 0.4829 | 0.3290 |
| Adapter [45] | 0.1086 | 0.1815 | 0.2345 | 0.2437 | 0.1211 | 0.177 | 0.1777 |
| WARP [40] | 0.1404 | 0.2556 | 0.3082 | 0.4369 | 0.1708 | 0.3684 | 0.2801 |
| IDPG [102] | 0.2596 | 0.2503 | 0.3334 | 0.4048 | 0.1984 | 0.3577 | 0.3007 |
| InfoPrompt | 0.2119 | 0.2993 | 0.3331 | 0.4739 | 0.2113 | 0.4034 | 0.3222 |
| $\gamma = 0$ | 0.2026 | 0.2834 | 0.3225 | 0.4776 | 0.2153 | 0.3739 | 0.3126 |
| $\beta = 0$ | 0.2013 | 0.2874 | 0.3208 | 0.4615 | 0.2072 | 0.3629 | 0.3069 |
| $\gamma = 0, \beta = 0$ | 0.1974 | 0.2728 | 0.3142 | 0.4662 | 0.2278 | 0.3276 | 0.3010 |

# 5 Experimental Results

## 5.1 Training with the Full dataset

Table 1 and 2 show the results of training with the full dataset for each task. We can observe that the results with our InfoPrompt are generally higher than those of the other parameters-efficient baselines that freeze the pretrained RoBERTa-Large parameters (*e.g.*, WARP and Adapter). Finetuning generally has better performance than the other approaches. This is because it allows training with all the model parameters, which is at the expense of more computation cost during training. As mentioned above Finetuning is intended to be included as the upper bound for performance. Moreover, we can find that the performance with $\gamma = 0$ and $\beta = 0$ is lower than that of InfoPrompt, indicating shows that it is beneficial to learn task-relevant prompt tokens with the proposed head loss and representation loss. Further, the performance gap between $\gamma = 0/\beta = 0$ and $\beta = \gamma = 0$ shows that the proposed functions are effective when added to naive prompt tuning, *i.e.*, with only $\mathcal{L}_{pred}$.

## 5.2 Training with the Few-Shot Datasets

The results for training with few-shot datasets are listed in Table 3 and 4. Compared with training with the full dataset (Table 1 and 2), we can find that the performance gap between our proposed InfoPrompt and the baselines is generally larger in the few-shot setting. Unlike the full datasets, the few-shot datasets contain much less information regarding the task to be learnt. As the result, the prompts learnt with solely the task loss (*e.g.*, with WARP or $\beta = \gamma = 0$) may easily overfit to the task-irrelevant information given the few-shot datasets. In such a scenario, it would be important to explicitly encourage the learnt prompt to be task-relevant, *i.e.*, via our proposed loss functions based on mutual information maximization. This explains why InfoPrompt yields larger performance gain when trained with few-shot datasets. Similar to training with the full datasets, the performance gains of InfoPrompt compared with InfoPrompt ($\gamma = 0$) and InfoPrompt ($\beta = 0$) show the effectiveness of our proposed loss functions in the few-shot scenario.

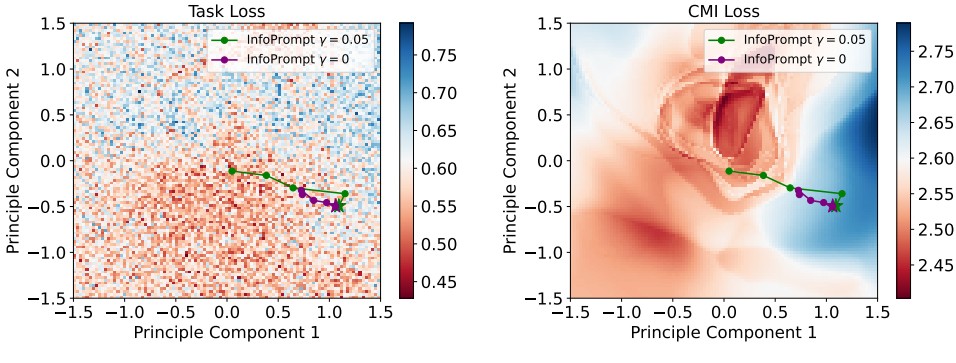

(a) The landscape of task loss. $\beta = 0.1$      (b) The landscape of representation loss. $\beta = 0.1$

Figure 2: The landscapes of the loss functions in the parameter space of the prompt tokens. The landscapes illustrates how the values of loss functions varies with the input prompt tokens. The trajectory shows the first 500 steps during training for InfoPrompt with $\gamma = 0.05$ or $\gamma = 0$.

## 6 Analysis

### 6.1 Loss Landscapes

To provide a more comprehensive understanding of the effectiveness of our proposed loss functions, we plot the landscapes of the loss functions in the parameter space of the prompt tokens. The landscapes illustrate how the values of the loss functions vary with the input prompt tokens. Since the prompt tokens are high-dimensional vectors, *i.e.,* each token has the dimension of 1024 for RoBERTa-Large, we visualize their associated loss values via projecting the prompt tokens into a 2D subspace. Specifically, we follow previous work on token embedding analysis [12] that projects the prompt tokens into the top-2 principal components computed from the pretrained token embeddings of RoBERTa-Large. We only insert one prompt token into the input sequence during visualization.

Taking the task of MRPC as an example, we plot the 2D landscapes of the task loss and the representation loss in Figure 2a and 2b, respectively. Both figures are plotted with the same scale, *i.e.*, with the same values of the prompt token. The axis values are the offset from the mean of the pretrained RoBERTa-Large token embeddings. The loss values shown in the figures are the average of 20 random samples from MRPC. In Figure 2a, we can find that there are a lot of local minimum in the landscapes of the task loss. This is consistent with the observations of the previous works [37, 91] that prompt tuning is difficult to be optimized with and sensitive to initialization, *e.g.*, the optimization can get easily overfit to a local minimum without proper initialization. From Figure 2b, we can observe that the loss landscape of our proposed representation loss is much smoother compared to the task loss in Figure 2a. With smoother landscapes, the optimization with our proposed loss functions can be more stable (also shown in Section 6.2), *i.e.,* less likely to be trapped in a local minimum and also guaranteed to converge according to our theoretical results (see Theorem 3.1). Additionally, we plot the trajectory of the first 500 steps during training for InfoPrompt ($\gamma = 0.05$) (green) and $\gamma = 0$ (purple) in Figure 2a and 2b. The stars in the plot indicate the initial value of the prompt before training. We find that training with $\gamma = 0.05$ can render a larger descent for both the task loss and representation loss, compared to $\gamma = 0$. As analyzed in Section 3.1, the language head is easier to learn than the prompt. As the result, parameters of the language head may contain more task-relevant information during the earlier stage of training. By maximizing the mutual information between the head parameter and prompt via the proposed head loss (weighted by $\gamma$), we encourage the learnt prompt to capture more task-relevant information in the initial training steps, thus resulting $\gamma = 0.05$ to have a larger descent than $\gamma = 0$ in the trajectories shown in Figure 2a and 2b. We also compare our initialization to some common initialization approaches: Random Uniform and Sampled Vocabulary [60, 37]. By Random Uniform, we randomly sample prompt initialization from the continuous latent space. By Sampled Vocabulary, we randomly sample prompt initialization from language model's vocabulary set. The final results by WARP (Random Uniform), WARP (Sampled Vocabulary) and InfoPrompt are 0.626, 0.672 and 0.706 respectively. The results

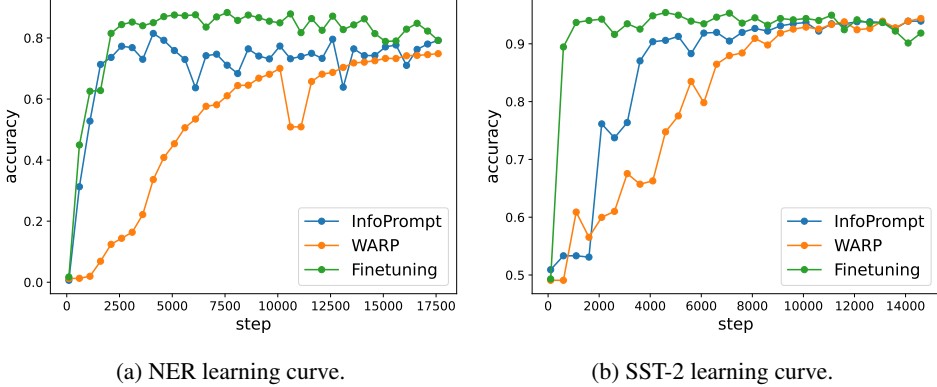

(a) NER learning curve.        (b) SST-2 learning curve.

Figure 3: The learning curves for the task of NER and SST-2. The training of our our InfoPrompt is more stabilized and converges faster, compared with WARP.

validate the effectiveness of our initialization approach. Note that our proposed two loss functions are unsupervised and do not require additional labels.

## 6.2 Learning Curve

We plot the training curve for the task of NER and SST-2 in Figure 3a and 3b, respectively. Unlike WARP [40] and Finetuning that train with solely the task loss $L_{\text{pred}}$, our InfoPrompt also trains with the representation loss and head loss. We can observe that the training of our our InfoPrompt is more stabilized and converges faster, compared with WARP. This can be explained from the landscape plots in Section 6.1. Since the landscape of the task loss is not smooth (Figure 2a), the training curve of WARP may exhibit significant perturbation when the optimization overfits to a local minimum, *e.g.*, the 10000th step in Figure 3a. Comparably, our proposed InfoPrompt can smooth the optimization landscape, thus stabilizing the training and result in faster convergence, which is guaranteed by our theoretical results. We observe that Finetuning generally converges faster and ends up with a higher accuracy than InfoPrompt. This is because Finetuning, which trains with all the model parameters, has much larger model capacity during training than prompt tuning (InfoPrompt and WARP). Such results for Finetuning is at the expense of larger computation costs, *i.e.*, we need to calculate the gradient for all the model parameters (354M) instead of only the prompt parameters $P$ (1.3M).

We also validate that our approach is less sensitive to initialization in the early learning stage, compared to WARP. Specifically, across 10 different random seeds, we report the standard errors of the performances by InfoPrompt and WARP in the early learning stage. After the first epoch, on NER, the results by WARP and InfoPrompt are $0.461 \pm 0.038$ and $0.810 \pm 0.025$, respectively. On SST2, the results by WARP and InfoPrompt are $0.735 \pm 0.033$ and $0.764 \pm 0.027$, respectively. The results show that our method has lower standard errors and is less sensitive compared to WARP.

## 7 Related Work

### 7.1 Soft Prompt Tuning

Soft prompt tuning has become a new paradigm in NLP. Based on some large pretrained models (e.g., BERT [25], RoBERTa [67]), a relatively small number of trainable parameters can be added to the input, while the parameters of backbones are fixed. Many works have demonstrated the effectiveness of soft prompt tuning in a wide range of NLP downstream tasks [60, 40, 73, 65], especially in low-resource regions [78, 66, 37]. Some recent works also found the transferable power of soft prompts across domains [101, 91, 94], across language models [91] and for zero-shot generalization [105]. To further enhance the efficiency of soft prompt parameters and enable better generalization abilities, many works consider multi-task learning [6, 27, 94, 43], or multilingual [14, 50]. Some works also try to explore the prompt with prior knowledge encoded [48, 42, 13]. While most of the initial attempts of soft prompt tuning are not context-aware, some recent works suggest that

the soft prompt tokens should be conditioned on the input context. Hyperprompt [43] proposes a hyper-network structure to generate prompt tokens based on task indexes. [102] and [8] suggest some context-aware soft prompt generation methods. [64] proposes a structured soft prompt tuning method. BBT [92] targets the scenarios where the pre-trained model is not available locally (i.e., deployed online) and its back-propagation operation is not available.

## 7.2 Information-theoretic Methods in NLP

Information-theoretic methods are widely used in many NLP tasks [55, 99, 90, 51, 70]. [99] and [55] propose information-theoretic methods for text memorization. [70] suggests an information-theoretic method for dialogue. [51] views the multimodal NMT problem in an information-theoretic point of view. For model pretraining, [96] proposes Infobert to improve the robustness of the BERT model. INFOXLM [15] proposes a cross-lingual language model based on an information-theoretic framework. For fine-tuning, [69] proposes an information bottleneck model method for low-resource fine-tuning. [89] introduces an information-theoretic method to engineer discrete prompts.

## 7.3 Theoretical Attention Computation

Softmax is one of the major unit in the attention scheme of most recent NLP large language models. Computing the attention matrix faster is a practically interesting question [17, 97, 57]. Recently, a number of theoretical work have tried to study the softmax/attention unit from theoretical perspective. The softmax attention computation can be formally defined as follows: suppose we are given three matrices $Q \in \mathbb{R}^{n \times k}$ (the query), $K \in \mathbb{R}^{n \times k}$ (the key), and $V \in \mathbb{R}^{n \times k}$ (the value), the goal is to compute $\mathsf{Att}(Q, K, V) = D^{-1} \exp(QK^\top)V$ where the diagonal matrix $D$ is $\mathrm{diag}(\exp(QK^\top)\mathbf{1}_n)$. Here $K^\top$ denote the transpose of matrix $K$. The work of [106, 3] consider the static setting, and the work of [11] considers the dynamic setting. [3] proposed a tight algorithm for computing Att and provided a lower bound result based on the strong exponential time hypothesis. [4] provide the results for a more general tensor version of attention which capture the three tuples feature, but classical attention cannot [77]. The work [11] shows a tight positive result and a negative result. In [11], they provide an upper bound via lazy update techniques [18]. In [11], they also present a lower bound result which is based on the Hinted MV conjecture [10]. The work of [21] proposes two sparsification algorithm to compute attention matrix when the feature dimension $\gg$ the length of sentence. [35] shows how to provide a differentially private algorithm for computing attention matrix under differential privacy framework [30, 29]. [41] introduces a hyperattention method and presents an nearly linear time algorithm with provable guarantees. [56] studies the polynomial based attention scheme and shows that sketching techniques can help speeding up the attention computation.

## 8 Conclusion and Future Work

We revisit the limitations of soft prompt tuning in the initialization. We also empirically discover that conventional prompt tuning methods cannot learn sufficient task-relevant information from prompt tokens. We tackle these limitations from an information-theoretic perspective and propose an information-theoretic prompt tuning method InfoPrompt, with two novel loss functions. With extensive experiments, without any prior expert knowledge, InfoPrompt can significantly accelerate the convergence of the prompt tuning and achieve more accurate and robust performances than traditional prompt tuning methods.

Existing instruction-tuned LMs (*e.g.*, Llama 2 [93]) are generally not task-specific and future works may further consider to tune such models with task-specific information. To achieve this, combining soft prompt tuning and prompt engineering [104, 76], from an information-theoretical perspective, can be a promising approach. Another future direction could be to further generalize our method to generation tasks (*e.g.*, sequence generation). In addition to prompt learning, it is interesting to explore how to extend our approach to other parameter-efficient fine-tuning methods (*e.g.*, LoRA [47] and HyperFormer [26]).

## Acknowledgments and Disclosure of Funding

The authors would like to thank the anonymous reviewers for their insightful comments. During this research, Ricardo Henao and Rui Wang were supported by ONR N00014-18-1-2871-P00002-3.

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

# Appendix

**Roadmap.** In Section A, we provide a number of basic notations. In Section B, we provide several basic definitions and discuss some related work about previous theoretical softmax regression results. In Section C, we provide a complete proof for our major theoretical result in this paper. We present our final result in Section D.

# A   Preliminaries

For any positive integer $n$, we use $[n]$ to denote set $\{1, 2, \cdots, n\}$. For any function $f$, we use $\widetilde{O}(g)$ to denote $g \cdot \mathrm{poly}(\log g)$.

**Vector**   For a length-$n$ vector $z$, we use $\exp(z)$ to denote a length-$n$ vector that its $i$-th entry is $\exp(z_i)$.

For a length-$n$ vector $z$, we use $\|z\|_2$ to represent its $\ell_2$ norm, i.e., $\|z\|_2 := (\sum_{i=1}^{n} x_i^2)^{1/2}$. For a length-$n$ vector $z$, we use $\|z\|_\infty$ to denote $\max_{i \in [n]} |z_i|$.

For a length-$n$ vector $z \in \mathbb{R}^n$, we use $\mathrm{diag}(z)$ to generate a diagonal matrix where each entry on the $(i, i)$-diagonal is $z_i$ for every $i \in [n]$.

We use $\mathbf{1}_n$ to represent a length-$n$ vector where all the coordinates are ones. Similarly, we use $\mathbf{0}_n$ to represent a length-$n$ vector where all the values are zeros.

**PSD**   We say $W \succeq Z$ (positive semi-definite) if $x^\top W x \geq x^\top Z x$ for all vector $x$.

**Matrix Related**   For an $n$ by $d$ matrix $C$, we use $\mathrm{nnz}(C)$ to denote the number of non-zero entries of $C$, i.e., $\mathrm{nnz}(C) := |\{(i, j) \in [n] \times [d] \mid C_{i,j} \neq 0\}|$

For a diagonal matrix $D \in \mathbb{R}^{n \times n}$, we say $D$ is a $k$-sparse diagonal matrix, i.e., $k = |\{i \in [n] \mid D_{i,i} \neq 0\}|$.

For any matrix $Z \in \mathbb{R}^{n \times k}$, we denote the spectral norm of $Z$ by $\|Z\|$, i.e.,

$$\|Z\| := \sup_{x \in \mathbb{R}^k} \frac{\|Zx\|_2}{\|x\|_2}.$$

For a matrix $Q$, we use $\sigma_{\max}(Q)$ to denote the largest singular value of $Q$. We use $\sigma_{\min}(Q)$ to denote the smallest singular value of $Q$.

**Matrix Computation**   We use $\omega$ to denote the exponent of matrix multiplication, i.e., $n^\omega$ denotes the time of multiplying an $n \times n$ matrix with another $n \times n$ matrix. Currently $\omega \approx 2.373$ [100, 58, 5].

**Calculus Related**   We use $\circ$ notation by following the literature's [63, 20, 36, 61, 80]. Suppose that we're given two column vectors $x, y \in \mathbb{R}^n$, we use $x \circ y$ to denote a column vector that $(x \circ y)_i$ is $x_i y_i$.

# B   Related Work about Theoretical Attention Regression Results

In this paper, we focus on the direction of regression tasks [32, 63, 20, 61, 80, 34, 23, 16, 33, 82, 24]. The goal of this section will review the linear regression (Definition B.1), exponential regression (Definition B.2), rescaled softmax regression (Definition B.3), softmax regression (Definition B.2).

**Definition B.1** (Linear regression). *Given a matrix $A \in \mathbb{R}^{n \times d}$ and $b \in \mathbb{R}^n$, the goal is to solve*

$$\min_{x \in \mathbb{R}^d} \|Ax - b\|_2.$$

*For convenient, let us $u(x)$ to denote $\exp(Ax)$.*

**Definition B.2** (Exponential Regression, see [32, 63]). *Suppose we are given a length $n$ vector $b$, and an $n$ by $d$ size matrix $A$, our goal is to optimize*

$$\min_{x \in \mathbb{R}^d} \|u(x) - b\|_2.$$

**Definition B.3** (Rescaled Softmax Regression, see [36]). *Suppose we are given a length $n$ vector $b$, and an $n$ by $d$ size matrix $A$, our goal is to optimize*

$$\min_{x \in \mathbb{R}^d} \|u(x) - \langle u(x), \mathbf{1}_n \rangle \cdot b\|_2$$

**Definition B.4** (Softmax Regression, see [20, 61, 80]). *Suppose we are given a length $n$ vector $b$, and an $n$ by $d$ size matrix $A$, our goal is to optimize*

$$\min_{x \in \mathbb{R}^d} \|\langle u(x), \mathbf{1}_n \rangle^{-1} \cdot u(x) - b\|_2.$$

## C  Theoretical Guarantees

In Section C.1, we provide several basic definitions. In Section C.2, we explain how to compute the gradient of function $f$. In Section C.3, we show how to compute the gradient of function $\log f(x)$. In Section C.4, we explain how to compute the Hessian of function $\log f(x)$. In Section C.5, we compute the hessian of inner product between $\log f(x)$ and $b$. In Section C.6, we compute the Hessian of cross entropy loss function. In Section C.7, we show Hessian is positive definite. In Section C.8, we prove that Hessian is Lipschitz. We remark that our experiments are based on first order method, and our theoretical proofs are mainly focusing on second order method. We believe it's an interesting future diretion to further study the convergence of the first order method such as [108, 1, 2, 88, 84, 83].

### C.1  Function Definition

We define

**Definition C.1.** *We define $u(x)$ as follows*

- $u(x) = \exp(Ax)$

**Definition C.2.** *We define $v(x)$ as follows*

- $v(x) = \exp(Ax)$

Previous [63] studies three types of hyperbolic functions $\exp(\cdot)$, $\cosh(\cdot)$ and $\sinh(\cdot)$. We mainly focus on $\exp(\cdot)$ function.

**Definition C.3** (Normalized coefficients, Definition 5.4 in [20]). *We define $\alpha : \mathbb{R}^d \to \mathbb{R}$ as follows*

$$\alpha(x) := \langle u(x), \mathbf{1}_n \rangle.$$

We define function softmax $f$ as follows

**Definition C.4** (Function $f$, Definition 5.1 in [20]). *Suppose that we're given an $n \times d$ matrix $A$. Let $\mathbf{1}_n$ denote a length-$n$ vector that all the coordinates are ones. We define prediction function $f : \mathbb{R}^d \to \mathbb{R}^n$ as follows*

$$f(x) := \langle u(x), \mathbf{1}_n \rangle^{-1} \cdot u(x).$$

**Fact C.5.** *Let $f(x)$ be defined as Definition C.4. Then we have*

- *Part 1. $\langle f(x), \mathbf{1}_n \rangle = 1$*

- *Part 2. $\|f(x)\|_1 = 1$*

- *Part 3. $\|f(x)\|_2 \leq 1$*

*Proof.* The proof is straightforward. For more details, we refer the readers to [20]. $\qquad\square$

We define the $\ell_2$ loss

**Definition C.6.** *We define*

$$L_{\exp} := 0.5\|f(x) - b\|_2^2.$$

Previous work [80] only considers entropy, here we consider cross entropy instead.

**Definition C.7** (Cross Entropy). *We define $L_{\mathrm{cent}} : \mathbb{R}^d \to \mathbb{R}$,*

$$L_{\mathrm{cent}}(x) := -\langle b, \log f(x) \rangle$$

**Definition C.8.** *Suppose we're given an $n \times d$ matrix $A$ and $W = \mathrm{diag}(w) \in \mathbb{R}^{n \times n}$ where $w \in \mathbb{R}^n$ is a vector, we define $L_{\mathrm{reg}} : \mathbb{R}^d \to \mathbb{R}$*

$$L_{\mathrm{reg}}(x) := 0.5\|WAx\|_2^2$$

## C.2 Gradient Computation for Function $f$

We present a calculus tool from previous work [20] (for example, we refer the readers to Lemma 5.6 in [20]).

**Lemma C.9.** *If the following conditions hold*

- *Given matrix $A \in \mathbb{R}^{n \times d}$ and a vector $b \in \mathbb{R}^n$.*
- *Suppose that function $\alpha : \mathbb{R}^d \to \mathbb{R}$ be defined in Definition C.3.*
- *Suppose that function $f : \mathbb{R}^d \to \mathbb{R}^n$ be defined in Definition C.4.*

*For each $i \in [d]$, we have*

- *Part 1.*

$$\frac{\mathrm{d}f(x)}{\mathrm{d}x_i} = -\langle f(x), A_{*,i} \rangle \cdot f(x) + f(x) \circ A_{*,i}$$

- *Part 2.*

$$\langle \frac{\mathrm{d}f(x)}{\mathrm{d}x_i}, A_{*,i} \rangle = -\langle f(x), A_{*,i} \rangle^2 + \langle f(x), A_{*,i} \circ A_{*,i} \rangle$$

- *Part 3.*

$$\langle \frac{\mathrm{d}f(x)}{\mathrm{d}x_i}, A_{*,j} \rangle = -\langle f(x), A_{*,i} \rangle \cdot \langle f(x), A_{*,j} \rangle + \langle f(x), A_{*,i} \circ A_{*,j} \rangle$$

## C.3 Gradient Computation for Function $\log f(x)$

In this section, we explain how to compute the gradient of $\log f(x)$.

**Lemma C.10.** *If the following condition holds*

- *Suppose that function $f$ is defined in Definition C.4.*

*We have*

- *Part 1.*

$$\frac{\mathrm{d}\log f(x)}{\mathrm{d}x_i} = -\langle f(x), A_{*,i} \rangle \cdot \mathbf{1}_n + A_{*,i}$$

- *Part 2.*

$$\langle \frac{\mathrm{d}\log f(x)}{\mathrm{d}x_i}, b \rangle = \langle A_{*,i}, b \rangle - \langle f(x), A_{*,i} \rangle \cdot \langle b, \mathbf{1}_n \rangle$$

- *Part 3.*

$$\frac{\mathrm{d}}{\mathrm{d}x_i} L_{\mathrm{cent}}(x) = \langle f(x), A_{*,i} \rangle \cdot \langle b, \mathbf{1}_n \rangle - \langle A_{*,i}, b \rangle$$

*Proof.* **Proof of Part 1.**

For all index $j \in [n]$, we can compute the gradient with respect to $x_i$

$$\frac{\mathrm{d} \log f(x)_j}{\mathrm{d}x_i} = f(x)_j^{-1} \frac{\mathrm{d}f(x)_j}{\mathrm{d}x_i}$$

Then we group the $n$ coordinates, we get

$$\begin{aligned}
\frac{\mathrm{d} \log f(x)}{\mathrm{d}x_i} &= f(x)^{-1} \circ \frac{\mathrm{d}f(x)}{\mathrm{d}x_i} \\
&= f(x)^{-1} \circ (-\langle f(x), A_{*,i} \rangle f(x) + f(x) \circ A_{*,i}) \\
&= -\langle f(x), A_{*,i} \rangle f(x)^{-1} \circ f(x) + f(x)^{-1} \circ f(x) \circ A_{*,i} \\
&= -\langle f(x), A_{*,i} \rangle \cdot \mathbf{1}_n + A_{*,i}
\end{aligned}$$

**Proof of Part 2.** We have

$$\begin{aligned}
\langle \frac{\mathrm{d} \log f(x)}{\mathrm{d}x_i}, b \rangle &= \langle -\langle f(x), A_{*,i} \rangle \cdot \mathbf{1}_n + A_{*,i}, b \rangle \\
&= \langle A_{*,i}, b \rangle - \langle f(x), A_{*,i} \rangle \cdot \langle b, \mathbf{1}_n \rangle,
\end{aligned}$$

where the first step follows from Part 1 and the second step follows from simple algebra.

**Proof of Part 3.** The proof directly follows from Part 2 and Definition of $L_{\mathrm{cent}}(x)$ (See Definition C.7). $\qquad \square$

### C.4   Hessian Computation for Function $\log f(x)$

In this section, we will show how to compute the Hessian for function $\log f(x)$.

**Lemma C.11.** *If the following conditions hold*

- *Let $f$ be defined as Definition C.4.*

*Then we have*

- *Part 1.*

$$\frac{\mathrm{d}^2 \log f(x)}{\mathrm{d}x_i^2} = (\langle f(x), A_{*,i} \rangle^2 - \langle f(x), A_{*,i} \circ A_{*,i} \rangle) \cdot \mathbf{1}_n$$

- *Part 2.*

$$\frac{\mathrm{d}^2 \log f(x)}{\mathrm{d}x_i \mathrm{d}x_j} = (\langle f(x), A_{*,i} \rangle \langle f(x), A_{*,j} \rangle - \langle f(x), A_{*,i} \circ A_{*,j} \rangle) \cdot \mathbf{1}_n$$

*Proof.* **Proof of Part 1.**

We have

$$\begin{aligned}
\frac{\mathrm{d}^2 \log f(x)}{\mathrm{d}x_i^2} &= \frac{\mathrm{d}}{\mathrm{d}x_i} \Big( \frac{\mathrm{d} \log f(x)}{\mathrm{d}x_i} \Big) \\
&= \frac{\mathrm{d}}{\mathrm{d}x_i} (-\langle f(x), A_{*,i} \rangle \cdot \mathbf{1}_n + A_{*,i}) \\
&= -\frac{\mathrm{d}}{\mathrm{d}x_i} (\langle f(x), A_{*,i} \rangle) \cdot \mathbf{1}_n \\
&= (\langle f(x), A_{*,i} \rangle^2 - \langle f(x), A_{*,i} \circ A_{*,i} \rangle) \cdot \mathbf{1}_n
\end{aligned}$$

where the 2nd step comes from Part 1 of Lemma C.10, the 3rd step follows from $A_{*,i}$ is independent of $x$, and the forth step follows from Part 2 of Lemma C.9.

**Proof of Part 2.**

Similarly, we can provide a proof for Part 2. $\qquad \square$

## C.5 Hessian Computation for Function $\langle \log f(x), b \rangle$

The goal of this section is to prove Lemma C.12.

**Lemma C.12.** *If the following conditions hold*

- *Let $f$ be defined as Definition C.4.*

*Then we have*

- *Part 1.*

$$\langle \frac{\mathrm{d}^2 \log f(x)}{\mathrm{d}x_i^2}, b \rangle = (\langle f(x), A_{*,i} \rangle^2 - \langle f(x), A_{*,i} \circ A_{*,i} \rangle) \cdot \langle \mathbf{1}_n, b \rangle$$

- *Part 2.*

$$\langle \frac{\mathrm{d}^2 \log f(x)}{\mathrm{d}x_i \mathrm{d}x_j}, b \rangle = (\langle f(x), A_{*,i} \rangle \langle f(x), A_{*,j} \rangle - \langle f(x), A_{*,i} \circ A_{*,j} \rangle) \cdot \langle \mathbf{1}_n, b \rangle$$

*Proof.* The proof directly follows from Lemma C.11. $\qquad \square$

## C.6 Hessian Computation for Function $L_{\mathrm{cent}}(x)$

For convenient of analyzing the $d \times d$ Hessian matrix, we will start with defining $n \times n$ matrix $B$.

**Definition C.13.** *We define $B(x) \in \mathbb{R}^{n \times n}$ as follows*

$$B(x) := \langle \mathbf{1}_n, b \rangle \cdot (\mathrm{diag}(f(x)) - f(x)f(x)^\top)$$

**Lemma C.14.** *If the following conditions hold*

- *Let $f$ be defined as Definition C.4.*
- *Let $L_{\mathrm{cent}}$ be defined as Definition C.7*
- *Let $B$ be defined as Definition C.13*

*Then we have*

- *Part 1.*

$$\frac{\mathrm{d}^2}{\mathrm{d}x_i^2} L_{\mathrm{cent}} = (-\langle f(x), A_{*,i} \rangle^2 + \langle f(x), A_{*,i} \circ A_{*,i} \rangle) \cdot \langle \mathbf{1}_n, b \rangle$$

- *Part 2.*

$$\frac{\mathrm{d}^2}{\mathrm{d}x_i \mathrm{d}x_j} L_{\mathrm{cent}} = (-\langle f(x), A_{*,i} \rangle \langle f(x), A_{*,j} \rangle + \langle f(x), A_{*,i} \circ A_{*,j} \rangle) \cdot \langle \mathbf{1}_n, b \rangle$$

- *Part 3.*

$$\frac{\mathrm{d}^2}{\mathrm{d}x^2} L_{\mathrm{cent}} = A^\top B(x) A$$

*Proof.* The proof trivially follows from Lemma C.12 and Definition C.13. $\qquad \square$

## C.7 Hessian is Positive Definite

Previous work [20] doesn't consider cross entropy into the final loss function. Here we generalize previous lemma so that cross entropy is also being considered.

**Lemma C.15** (A cross entropy generalization of Lemma 6.3 in [20])**.** *Suppose the following conditions hold*

- *Let $A \in \mathbb{R}^{n \times d}$, $R \geq 4$, $l > 0$, suppose that $R_0 = \exp(O(R^2 + \log n))$*

- 

$$L(x) = \underbrace{L_{\mathrm{reg}}(x)}_{\text{Definition C.8}} + \underbrace{L_{\mathrm{cent}}(x)}_{\text{Definition C.7}} + \underbrace{L_{\mathrm{exp}}(x)}_{\text{Definition C.6}} \ .$$

- *Let $\widetilde{B}(x) = B(x) + W^2$*

*Then we have*

- *Part 1. $\min_{i \in [n]} w_i^2 \geq 10 R_0 + l/\sigma_{\min}(A)^2$, then we have*

$$\frac{\mathrm{d}^2 L}{\mathrm{d}x^2} \succeq l \cdot I_d$$

- *Part 2. $\min_{i \in [n]} w_i^2 \geq 10^4 \cdot R_0 + l/\sigma_{\min}(A)^2$, then we have*

$$(1 - 0.01) \cdot \widetilde{B}(x) \preceq W^2 \preceq (1 - 0.01) \cdot \widetilde{B}(x).$$

*Proof.* Using the definition of $B$ for $L_{\mathrm{cent}}$ (see Definition C.13), definition/bound of $B$ for $L_{\mathrm{exp}}$ (see [20]), and tools developed in Section 6 in [20], we can finish the proof. □

### C.8 Hessian is Lipschitz

Previous work [20] doesn't consider cross entropy into the final loss function. Here we generalize previous lemma so that cross entropy is also being considered.

**Lemma C.16** (A cross entropy version of Lemma 7.1 in [20]). *Suppose the following condition holds*

- *Let $H(x) = \frac{\mathrm{d}^2 L}{\mathrm{d}x^2}$ and $R > 4$*

- *Suppose that $\max\{\|x\|_2, \|y\|_2\} \leq R$, and $\max\{\|A\|, \|b\|_2\} \leq R$*

- *$\|A(x - y)\|_\infty < 0.01$*

*Then we have*

$$\|H(x) - H(y)\| \leq n^4 \exp(O(R^2 + \log n)) \cdot \|x - y\|_2$$

*Proof.* Using the definition of $B$ for $L_{\mathrm{cent}}$ (see Definition C.13), definition/bound of $B$ for $L_{\mathrm{exp}}$ (see [20]), and tools developed in Section 7 in [20], we can finish the proof. □

---

**Algorithm 1** Our Algorithm.

---

1: **procedure** OURALGORITHM($A \in \mathbb{R}^{n \times d}, b \in \mathbb{R}^n, w \in \mathbb{R}^n, \epsilon, \delta$)  ▷ Theorem D.1
2:   We choose $x_0$
3:   $T \leftarrow \log(\|x_0 - x^*\|_2/\epsilon)$  ▷ $T$ denotes the number of iterations
4:   **for** $t = 0 \rightarrow T$ **do**
5:     Implicitly formulate exact Hessian and use that to construct an approximate Hessian $\widetilde{H}$ (similar as Section 8 in [20])
6:     Compute gradient
7:     $\widetilde{H} \leftarrow A^\top \widetilde{D} A$
8:     $x_{t+1} \leftarrow x_t + \widetilde{H}^{-1} g$
9:   **end for**
10:   $\widetilde{x} \leftarrow x_{T+1}$
11:   **return** $\widetilde{x}$
12: **end procedure**

---

# D  Main Theoretical Guarantees

Previous work [20] has proved the similar result without considering the cross entropy. We generalize the techniques in previous paper [20] from only considering $\ell_2$ task loss to considering both $\ell_2$ task loss and cross entropy loss ($L_{\text{cent}}$ see formal definition in Definition C.7). Our algorithm is a version of approximate Newton method, such methods have been widely used in many optimization tasks [18, 59, 9, 52, 87, 54, 38, 85, 49, 74, 20, 39, 53, 86]. In this work, we focus on the approximate Newton method along the line of [85, 22, 20].

**Theorem D.1** (Formal version of Theorem 3.1). *Let $x^*$ denote an length-$d$ vector that is satisfying,*

$$\arg\min_{x \in \mathbb{R}^d} \underbrace{L_{\text{exp}}}_{\text{Definition C.6}} + \underbrace{L_{\text{cent}}}_{\text{Definition C.7}} + \underbrace{L_{\text{reg}}}_{\text{Definition C.8}}$$

*Suppose the following conditions are holding:*

- $R \geq 4$, $g(x^*) = \mathbf{0}_d$.
- $\|x^*\|_2 \leq R$, $\|A\| \leq R$, $\|b\|_2 \leq R$.
- $M = \exp(O(R^2 + \log n))$.
- $\min_{i \in [n]} w_i^2 \geq 100M + l/\sigma_{\min}(A)^2$
- *Suppose that $\epsilon \in (0, 0.1)$ is the final and $\delta \in (0, 0.1)$ is the failure probability.*
- *Suppose $x_0$ satisfy condition $M\|x_0 - x^*\|_2 \leq 0.1l$.*
- *Suppose that $T = \log(\|x_0 - x^*\|_2/\epsilon)$*

*Then there is a randomized algorithm (Algorithm 1) such that*

- *it runs $T$ iterations*
- *in each iteration, it spends time[‡]*

$$O((\text{nnz}(A) + d^\omega) \cdot \text{poly}(\log(n/\delta))).$$

- *generates a vector $\widetilde{x} \in \mathbb{R}^d$ that is satisfying*

$$\|\widetilde{x} - x^*\|_2 \leq \epsilon$$

- *the succeed probability is $1 - \delta$*

*Proof.* The high level framework of our theorem is similar to previous work about exponential regression [63], softmax regression [20] and rescaled softmax regression [36]. Similarly as previous work [63, 20, 36, 80], we use the approximate newton algorithm (for example see Section 8 in [20]). So in the proof, we only focus on the difference about the Hessian positive definite lower bound and Hessian Lispchitz property.

Using Lemma C.15 and Lemma C.16 and approximate Newton algorithm analysis in [20], then we complete the proof. ∎

# E  Comparison with Parameter-Efficient Fine-tuning Methods Not Based on Prompt Tuning

In this section, we focus on the comparison between InfoPrompt ($N_p = 4$) and some PEFT (Parameter-Efficient Fine-tuning Methods) baselines which are not based on prompt tuning:

- Adapter [45]: Similar to prompt tuning, this is also a way of parameter-efficient training for pretrained language models. Specifically, instead of adding the prompt tokens in the input, we add adapters after the feed-forward module in each transformer layer.

---

[‡]Here $\omega$ denotes the exponent of matrix multiplication. Currently $\omega \approx 2.373$.

Table 5: Comparison with parameter-efficient fine-tuning methods which are not based on prompt tuning. The number of prompt is fixed to $n_p = 4$ for the soft prompt tuning method.

| Full datasets | CoLA | RTE | MRPC | SST2 | RE | NER | SemEval | Average |
|---|---|---|---|---|---|---|---|---|
| LoRA | 0.5880 | 0.6715 | 0.8235 | 0.9541 | 0.6636 | 0.8228 | 0.7214 | 0.7492 |
| Adapter | 0.5552 | 0.5776 | 0.6814 | 0.9472 | 0.5073 | 0.8329 | 0.6570 | 0.6798 |
| InfoPrompt | 0.6018 | 0.6968 | 0.8137 | 0.9599 | 0.7616 | 0.8962 | 0.7917 | 0.7888 |
| $N = 64$ | CoLA | RTE | MRPC | SST2 | RE | NER | SemEval | Average |
| LoRA | 0.0991 | 0.5596 | 0.6985 | 0.5677 | 0.1232 | 0.1345 | 0.1711 | 0.3362 |
| Adapter | 0.0627 | 0.5487 | 0.5931 | 0.4908 | 0.1086 | 0.2345 | 0.1211 | 0.3085 |
| InfoPrompt | 0.1567 | 0.6137 | 0.7059 | 0.6697 | 0.2119 | 0.3331 | 0.2113 | 0.4146 |
| $N = 256$ | CoLA | RTE | MRPC | SST2 | RE | NER | SemEval | Average |
| LoRA | 0.2854 | 0.5740 | 0.7206 | 0.8222 | 0.2291 | 0.1955 | 0.3817 | 0.4583 |
| Adapter | 0.2486 | 0.5668 | 0.6250 | 0.6640 | 0.1815 | 0.2437 | 0.1770 | 0.3866 |
| InfoPrompt | 0.1750 | 0.6580 | 0.7377 | 0.7305 | 0.2993 | 0.4739 | 0.4034 | 0.4968 |

- LoRA [47]: Another parameter-efficient training method for pretrained language models. Specifically, LoRA adds additional low-rank decomposed matrices into each Transformer layer via residual connections.

In Table 5, we can observe that LoRA is a stronger baseline than Adapter. However, our method can still outperform LoRA, especially in the downstream tasks which require more task-relevant information, *e.g.*, NER and RE.

