# OpenReview forum: "InfoPrompt: Information-Theoretic Soft Prompt Tuning for Natural Language Understanding"
_NeurIPS.cc/2023/Conference — NeurIPS 2023 poster_

### Official Review · Reviewer_ysu9 · 2023-07-03

**Soundness:** 4 excellent
**Presentation:** 4 excellent
**Contribution:** 3 good
**Rating:** 7
**Confidence:** 3

**Summary:**

The authors formulate soft-prompt tuning as maximising mutual information between prompts and model parameters/encoded representations. The prompt is trained to maximise mutual information with the language model head parameters, and the representation of a task encoding from the model. The authors primarily test on GLUE, relation extraction, and NER tasks, and find that their approach closes the gap the most with full-finetuning. Analysis suggests that the proposed method is more stable and converges faster during training, perhaps due to the smoother loss landscape of the info loss.


**Strengths:**

Interesting approach, with clear motivation and description. The experiments are thorough, over a reasonable number of tasks, and show clear improvements over baselines. The analysis also provides useful and interesting insights into how the approach works, and improving both performance and convergence speed of soft prompt tuning is a useful and important result.

**Weaknesses:**

No glaring weaknesses, but I think the following could be improved:
- Comparisons with popular parameter-efficient finetuning (PEFT) methods: soft prompt tuning is often considered one of several peft methods, including LoRA. While you do compare with adapters, it would be interesting to compare against Lora or MaM adapters [1], which are generally considered more effective than adapters.
- The work claims their approach makes prompt tuning less sensitive to initialization. It would be good to see results over multiple seeds (or even multiple initialisation types), with smaller variation than a baseline approach, to back up this claim.

Overall, I think this is solid work and lean to accept it.

[1] He et al., 2022. *Towards a Unified View of Parameter-Efficient Transfer Learning*. https://openreview.net/pdf?id=0RDcd5Axok


**Questions:**

- How does InfoPrompt compare to lora (and other PEFT methods)?
- Could you apply a similar information loss technique to lora or other PEFT methods, and how effective would that be?
- How did you choose the values for beta and gamma?


**Limitations:**

The authors discuss limitations at the end of the paper to a reasonable degree.

---

> ### Author Rebuttal · Authors · 2023-08-10
>
> Thanks for your valuable comments!
>
> Weakness 1 & Q1 & Q2: ***Comparisons with popular parameter-efficient finetuning (PEFT) methods***
>
> Response: Thanks for your valuable suggestion. Below, we have introduced the Info-LoRA, which outperforms several other PEFT methods (e.g., adapter[4], BitFit[5]).
> Info-LoRA is implemented by combining LoRA finetuning and our soft-prompt tuning method. In addition to the soft-prompt tokens, we introduce additional parameters by LoRA and these parameters are also tunable in our downstream tasks.
>
> | Full-data | CoLA | RTE | MRPC | SST2 | Average |
> |---|---|---|---|---|---|
> | info-LoRA | 0.683 | 0.8736 | 0.9069 | 0.9667 | 0.8575 |
> | LoRA | 0.5880 | 0.6715 | 0.8235 | 0.9541 | 0.7592 |
>
> | 64 shots | CoLA | RTE | MRPC | SST2 | Average |
> |---|---|---|---|---|---|
> | info-LoRA | 0.0988 | 0.5776 | 0.7328 | 0.5333 | 0.4856 |
> | LoRA | 0.0991 | 0.5596 | 0.6985 | 0.5677 | 0.4812 |
>
> | 256 shots | CoLA | RTE | MRPC | SST2 | Average |
> |---|---|---|---|---|---|
> | info-LoRA | 0.3783 | 0.6462 | 0.7941 | 0.7362 | 0.6387 |
> | LoRA | 0.2854 | 0.5740 | 0.7206 | 0.8222 | 0.6005 |
>
> Weakness 2: ***Sensitivity to initialization.***
>
> Response: Thanks for your comment on this. We have some related discussion in Section 6.1 (line 255). We also have additional results to validate that, our approach is less sensitive in the early learning stage compared to the baseline approaches.
> For the results in Figure 3, we further reported the standard errors in the early learning stage. Specifically, we used 10 different random initializations for our method and the baseline WARP, and then reported the standard errors of the performances in the early learning stage (e.g., after the first epoch). The results show that our method's standard errors are lower, compared to WARP.
>
> |  | SST2 | NER-ACE |
> |---|---|---|
> | WARP | 0.7352 $\pm$ 0.0334 | 0.4606 $\pm$ 0.0385 |
> | InfoPrompt | 0.7639 $\pm$ 0.0270 | 0.8099 $\pm$ 0.0253 |
>
>
> Q3: ***How did you choose the values for beta and gamma?***
>
> Response: Similar to previous works [1, 2], we empirically choose the current hyper-parameters (without using the validation data). Specifically, based on our observation of the scales of the three losses, we choose the current hyper-parameters (without using validation data), so the scales of the three loss functions after weighting (head loss and representation loss, and task loss) are similar in the final loss function. This strategy is simply based on the inherent assumption that each loss should contribute equally to the problem [3].
>
> [1] Jia, Zhiwei, and Hao Su. "Information-theoretic local minima characterization and regularization." International Conference on Machine Learning. PMLR, 2020.
>
> [2] Shi, Yufeng et al. “Information-Theoretic Hashing for Zero-Shot Cross-Modal Retrieval.” ArXiv abs/2209.12491 (2022): n. pag.
>
> [3] Groenendijk, Rick, et al. "Multi-loss weighting with coefficient of variations." Proceedings of the IEEE/CVF winter conference on applications of computer vision. 2021.
>
> [4] Houlsby, Neil, et al. "Parameter-efficient transfer learning for NLP." International Conference on Machine Learning. PMLR, 2019.
>
> [5] Zaken, Elad Ben, Shauli Ravfogel, and Yoav Goldberg. "Bitfit: Simple parameter-efficient fine-tuning for transformer-based masked language-models." arXiv preprint arXiv:2106.10199 (2021).

---

> > ### Comment · Reviewer_ysu9 · 2023-08-14
> > **Re: Rebuttal**
> >
> > Hi, thanks for the response and follow-up experiments! I've carefully read your rebuttal and the other reviews and am keeping my score. I think better methods for soft prompt tuning are interesting and useful for the field, and while this paper is somewhat incremental, the experiments and proposed approach appear solid and are useful for future researchers studying prompt tuning. Ideally, I agree with other reviewers that a wider range of experiments would be useful, especially generation tasks using a generation model (T5 or llama), but I think the paper is okay with the current set of experiments + the ones the authors have performed for rebuttal.
> >
> > I hope the authors release their code in the future to aid reproducibility.

---

> > > ### Author Response · Authors · 2023-08-21
> > >
> > > Thank you very much for the acknowledgement. We deeply appreciate your time and effort in the review. We will certainly release the code to the community, and add the experiments in the rebuttal into the final version of the paper.

---

### Official Review · Reviewer_7rsH · 2023-07-04

**Soundness:** 2 fair
**Presentation:** 2 fair
**Contribution:** 2 fair
**Rating:** 4
**Confidence:** 4

**Summary:**

This work focuses on the initialization inefficiency of soft prompt tuning. Crucially, soft prompt tuning is notoriously challenging to obtain a good initialization, and highly sensitive to some hyper-parameters, leading to some optimization difficulty especially in low-resource scenarios. In terms of this, this work starts from an information-theoretic perspective of maximizing mutual information to get a better initialization of soft prompts. The experiments on classification, relation extraction, and NER tasks show the effectiveness of their design to several baselines.

**Strengths:**

1. The information-theoretic guarantee with some clearly organized mathematical equations is correct, and interesting as their designs, inspired by some contrastive learning techniques.
2. The experiments do show the effectiveness of their proposed algorithm, mainly in advanced NLU tasks, e.g., NER, and RE.


**Weaknesses:**

1. Of course, this work has some contributions to soft prompt tuning initialization from a different perspective, even such perspective is highly similar to existing work on discrete prompt engineerings [1]. However, I think it has some limited impacts, as your task settings are somewhat less interesting given a lot of works on better discrete prompt optimization with significant better performance under few-shot setting, and many work simply initializing the soft prompts by pre-training on similar datasets show good performance as well. Moreover, current main focus in the community would not be how to tune such negligible soft prompts to a limited range of NLP tasks, instead, people would use better instruction-tuned LMs for real-world application, etc.

2. You need to re-polish your writing, as I do frequently find some typos, e.g., performances in page1, intro, "your method names -- Infopropmt" in section 9., limitation, and so on.

3. Your few-shot numbered improvements, which might be your main contributions in your introduction, are still negligible to some extent, further restricting your impacts in this conference.
[1] An Information-theoretic Approach to Prompt Engineering Without Ground Truth Labels, ACL 2022

**Questions:**

No.

**Limitations:**

Please see my reply on the weaknesses. Their listed limitations are too general as the typical limitation of prompt tuning...

---

> ### Author Rebuttal · Authors · 2023-08-10
>
> Thanks for your valuable comments!
>
> Weakness 1: ***your task settings are somewhat less interesting given a lot of works on better discrete prompt optimization with significant better performance under few-shot setting***
>
> Response:
> We would like to politely contend that the discrete and (vs.) soft prompt tuning (PT) are two parallel setups with pre-trained language models [2].
> As mentioned in Section 2 of [1] (mentioned in the review), the best prompts are tuned with the continuous embeddings space (soft PT).
> This is because the continuous the search space of the discrete PT (i.e., a discrete set of token embeddings) is actually a subset of the search space of soft PT (i.e. the whole embedding space).
> Thus, the soft continuous prompts should be more expressive with a large search space, and the performance of the optimal prompt from soft PT is expected to upper bound that from discrete PT [3] (Section 7.2).
>
> Additionally, it is computational challenging with discrete PT to optimize in a discrete search space [2,3]. Comparative, our proposed objectives of soft PT are optimized continuously with provable convergence guarantee (Theorem 1).
>
>
> Weakness 1: ***many work simply initializing the soft prompts by pre-training on similar datasets***
>
> Response:
> Pre-training on similar datasets indeed may lead to better performance.
> However, such a transfer learning set up is orthogonal to our problem formulation, i.e., we consider the scenario where no auxiliary datasets are available.
> Our scenario is actually more practical since it can be difficult and expensive to identify the similar datasets of the target task [7].
>
>
> Weakness 2: ***You need to re-polish your writing, as I do frequently find some typos***
>
> Response: Thanks for pointing out our typos. We will fix them accordingly.
>
>
> Weakness 3: ***Moreover, current main focus in the community would not be how to tune such negligible soft prompts......instead, people would use better instruction-tuned LMs for......***
>
> Response:
> Please allow us to respectfully disagree with this statement. Firstly, we believe that tuning soft prompts is hardly considered negligible. Actually, it is an active field of efficient training with pre-trained language models and is being widely studied by many recent papers, e.g., [4-7]. Secondly, existing instruction-tuned LMs (e.g., GPT-3.5/4) are generally not task-specific, thus still yet to excel when being evaluated on many of the NLP tasks [8,9]. Therefore, it is still necessary to further tune such models with task-specific information. For this purpose, soft prompt tuning is an efficient approach and is still being actively studied [2-7].
>
> Besides, although instruction-tuned LMs have been actively studied for real-world application, we have noticed some very recent works showing that instruction-tuned LMs and soft-prompt tuning are complementary. By properly developing soft-prompt tuning methods, it is promising to further improve the instruction-tuned LMs [10, 11].
>
> [1] An Information-theoretic Approach to Prompt Engineering Without Ground Truth Labels, ACL 2022
>
> [2] Liu, Pengfei, et al. "Pre-train, prompt, and predict: A systematic survey of prompting methods in natural language processing." ACM Computing Surveys 55.9 (2023): 1-35.
>
> [3] Li, X. L., & Liang, P. (2021, August). Prefix-Tuning: Optimizing Continuous Prompts for Generation. In Proceedings of the 59th Annual Meeting of the Association for Computational Linguistics and the 11th International Joint Conference on Natural Language Processing (Volume 1: Long Papers) (pp. 4582-4597).
>
> [4] Razdaibiedina, Anastasia, et al. "Progressive Prompts: Continual Learning for Language Models." The Eleventh International Conference on Learning Representations. 2022.
>
> [5] Wang, Zhen, et al. "Multitask Prompt Tuning Enables Parameter-Efficient Transfer Learning." The Eleventh International Conference on Learning Representations. 2022.
>
> [6] Razdaibiedina, Anastasia, et al. "Residual Prompt Tuning: Improving Prompt Tuning with Residual Reparameterization." arXiv e-prints (2023): arXiv-2305. (Findings of ACL2023)
>
> [7] Pang, Bo, et al. "SharPT: Shared Latent Space Prompt Tuning." Findings of the Association for Computational Linguistics: EACL 2023. 2023.
>
> [8] GPT-4 Technical Report, OpenAI.
>
> [9] Koubaa, Anis. "GPT-4 vs. GPT-3.5: A concise showdown." (2023).
>
> [10] Sun, Simeng, et al. "How does in-context learning help prompt tuning?." arXiv preprint arXiv:2302.11521 (2023).
>
> [11] Shi, Zhengxiang, and Aldo Lipani. "Don't Stop Pretraining? Make Prompt-based Fine-tuning Powerful Learner." arXiv preprint arXiv:2305.01711 (2023).

---

> > ### Comment · Reviewer_7rsH · 2023-08-16
> > **Replies from Reviewer 7rsH**
> >
> > Thanks for your reply! I am acknowledging your contributions to soft prompt tuning on three tasks (including classification, NER/RE) under full-shot and few-shot scenarios compared to traditional soft prompt tuning, but there does exist some potential limitations which deserve your future hard work to improve the contents for a more rigorous paper, instead of any possible over-claims.
> >
> > -- All these replies are to ask for more rigorous studies to position your contribution.
> >
> > Your main contribution: "A different soft prompt tuning framework based on mutual information, a different perspective which brings some new insights, and advantages, such as **initialization/efficiency (here, higher performance under low-resource scenarios), and effectiveness**". Let me know if I have any misunderstandings.
> >
> > So in order to support your current claims on your advantages, e.g., efficiency at better capturing task data knowledge, you shall conduct rich experiments to show your advantages (yes, you do evaluate on traditional prompt tuning baselines, but for advantages in initialization, it is necessary to form a fair comparison against other initialization baselines --- see my words later, and perhaps for NLU title, it is still somewhat over-claimed), **in addition to insights related to your central topic** -- "information theoretical soft prompt tuning".
> >
> > <1. Task settings>
> > - For the two options I listed here, including discrete prompt optimization and continuous prompt tuning, I think these two altogether limit your impacts in terms of the performance and the applicability at the beginning. It is just to say that it would be good for you to include their performance as well, to position your **applicability** on your currently tested tasks if you want to highlight this (Or if your selling point is your **low-resource initialization** (efficient in your abstract), it could be better to incorporate some baselines for this problem, like [7], which sets the baseline with SPoT-retrieval, but in terms of this top-tier conference, more baselines in this initialization point would definitely be a plus). -- But of course, this point shall be marginal, compared to your central topic/contribution: "a different starting point on mutual information".
> >
> > - And for NLU, if you want to claim your approach (add more values to your proposal) does contribute to a wide range of NLU tasks, you need to show more numbers on other task formats, e.g., QA task, and even more, since standard NLU is too broad with many possible prompt adaptations.  --- So to avoid any over-claims.
> >
> > - To claim your approach is better at few-shot scenarios, it would be good to provide some more initialization baselines like your [7]. Currently, you basically assume that in your current task formulation, you do not have any additional similar datasets/resources to use in your few-shot sections (low-resource in your [7]). In [7], to show their framework is better in initialization, they already incorporate baselines like SPoT-retrieval, where you lack some rigorous empirical evaluations w/ even heuristics-based methods. Instead, you just test with other common prompt tuning baselines to say your efficiency is significant. Yes, compared to your baselines, it is true. But how about its potential to the real progress on "prompt initialization"?
> >
> > <2. Soft Prompts ==> Marginal Numbered improvements>
> > - Sorry for the ambiguous language, which makes you **misunderstand my words**. I am not saying that soft prompts are negligible (please pay attention to my context carefully), but saying I am curious about whether your soft prompt tuning work with *marginal numbers (or limited studies w/ possible baselines) compared to other prompting paradigms, not just your current traditional prompt tuning baselines* could be significant to the field, instead of some incremental progress. For soft prompts, there are indeed many potential hot topics, such as prompt distillation/compression for text counterparts, same for the integrations between soft prompts and instruction-tuned LMs. Indeed, I am saying in your task scenarios, you have possible other alternatives deserving your empirical comparisons. And even more, other suitable soft tuning methods outperform yours (BBT, LPT, etc).
> > - Of course, I acknowledge your contributions of a different prompt tuning framework to soft prompt tuning.
> >
> > <3. one of your contributions>
> > - You have claimed that you also provide formal derivations of your information-theoretical framework in your pending appendix as one of your contributions. So I cannot provide one very decisive rating for this, which is not very objective and fair.
> >
> > <Minor>
> > - Some typos.
> >
> > In terms of the above reasons, I still think this paper needs further improvements, which is the main reason that I keep my current borderline ratings. It would be good for ACs to help on the final judgements after reading my comments.
> >
> > Thanks, and hope my elaborations could address your concerns.

---

> > > ### Author Response · Authors · 2023-08-21
> > >
> > > Thank you very much for the insightful comments to improve the position of our contribution. We appreciate your acknowledgement of our contribution to soft prompt tuning.
> > >
> > > <1. Task settings>
> > >
> > > ***discrete prompt optimization and continuous prompt tuning***
> > >
> > > We include the experiments with discrete prompt optimization listed below. Due to limited time of the rebuttal discussion period, we show four experiments with 64 shots. By MI-discrete, we follow [16] to design a set of candidate discrete prompts, and follow the mutual information metric in [16] to select the best discrete prompt. By MI-discrete+InfoPrompt, in addition to finding the best discrete prompt as in [16], we also insert the soft prompt and apply our soft prompt tuning method. We observe that approaches of soft prompt tuning (WARP and InfoPrompt) are better than MI-discrete, since soft prompt tuning has a larger search space of optimization than the discrete approaches. We also experiment with MI-discrete+InfoPrompt that achieves better performance than MI-discrete. This shows that the soft prompt tuning and discrete prompt tuning are not mutually exclusive. Instead, our propose soft InfoPrompt can be combined with the discrete prompt tuning to further improve MI-discrete's performance.
> > >
> > >
> > > |                          | RTE    | SST2   | MRPC   | CoLA   |
> > > |--------------------------|--------|--------|--------|--------|
> > > | WARP                     | 0.5596 | 0.5872 | 0.7083 | 0.0749 |
> > > | MI-discrete              | 0.5271 | 0.5344 | 0.6838 | 0.0547 |
> > > | | |
> > > | MI-discrete+InfoPrompt   | 0.5848 | 0.6812 | 0.7206 | 0.1152 |
> > > | InfoPrompt               | 0.6137 | 0.6697 | 0.7059 | 0.1567 |
> > >
> > >
> > > ***NLU tasks***
> > >
> > > Thank you for the suggestion to avoid any over-claims. Similar to previous works [12-16] claiming to contribute to improved methods in natural language understanding, we followed their evaluation protocol and evaluated our approach on GLUE, ACE, and SemEval, instead of evaluating our approach on the QA task.
> > >
> > > [12] Liu, Xiaodong, et al. "Multi-task deep neural networks for natural language understanding." arXiv preprint arXiv:1901.11504 (2019).
> > >
> > > [13] Clark, Kevin, et al. "Bam! born-again multi-task networks for natural language understanding." arXiv preprint arXiv:1907.04829 (2019).
> > >
> > > [14] Zhang, Zhuosheng, et al. "Semantics-aware BERT for language understanding." Proceedings of the AAAI Conference on Artificial Intelligence. Vol. 34. No. 05. 2020.
> > >
> > > [15] Zhang, Taolin, et al. "DKPLM: decomposable knowledge-enhanced pre-trained language model for natural language understanding." Proceedings of the AAAI Conference on Artificial Intelligence. Vol. 36. No. 10. 2022.
> > >
> > > [16] Sorensen, Taylor, et al. "An Information-theoretic Approach to Prompt Engineering Without Ground Truth Labels." Proceedings of the 60th Annual Meeting of the Association for Computational Linguistics (Volume 1: Long Papers). 2022.
> > >
> > >
> > >
> > > ***initialization baselines***
> > >
> > > Thank you for the insightful comments. The baseline approaches in [7] is orthogonal to our problem formulation due to its transfer learning setup, i.e., requires additional resources (datasets). In our experiments, we consider the scenario where no auxiliary datasets are available. Thus, we only include initialization baselines without additional resource requirements for fair comparison.
> > >
> > > To validate the effectiveness of our initialization approach, we compare our initialization to some common initialization approaches mentioned in [17, 18]. Specifically, in addition to the baseline WARP using class-label initialization in our paper, we further report the baselines with Random Uniform and Sampled Vocabulary initialization. By Random Uniform, we randomly sample prompt initialization from the continuous latent space.  By Sampled Vocabulary, we randomly sample prompt initialization from language model's vocabulary set. Due to the limited time of the rebuttal discussion period, we present the results on 4 datasets, and we will certainly include more results in the updated version.
> > >
> > > |                   | RTE    | SST2   | MRPC   | CoLA   |
> > > |-------------------|--------|--------|--------|--------|
> > > | WARP (Random Uniform)     | 0.5211 | 0.5391 | 0.6265 | 0.0312 |
> > > | WARP (Sampled Vocabulary) | 0.5475 | 0.6140 | 0.6725 | 0.0602 |
> > > | InfoPrompt        | 0.6137 | 0.6697 | 0.7059 | 0.1567 |
> > >
> > >
> > > [17] Lester, Brian, Rami Al-Rfou, and Noah Constant. "The power of scale for parameter-efficient prompt tuning." arXiv preprint arXiv:2104.08691 (2021).
> > >
> > > [18] Gu, Yuxian, et al. "Ppt: Pre-trained prompt tuning for few-shot learning." arXiv preprint arXiv:2109.04332 (2021).
> > >
> > > (***Please check our additional responses to the other questions in the next comment.***)

---

> > > > ### Author Response · Authors · 2023-08-21
> > > >
> > > > (***Please refer to the previous comment for our responses to the earlier questions.***)
> > > >
> > > > <2. Soft Prompts ==> Marginal Numbered improvements>
> > > >
> > > > To make our comparison with baselines more comprehensive, we have included an additional experiment with LPT [19] below. The results with LPT are following the configuration in the original paper [19]. The results with our proposed training objectives (InforPrompt) outperforms the baselines.
> > > >
> > > > |    | RTE| SST2  | MRPC  | CoLA  |
> > > > |------|------|-------|-------|-------|
> > > > | LPT  |0.5379|0.6560|0.6838|0.1010|
> > > > | WARP |0.5596|0.5872|0.7083|0.0749|
> > > > | InfoPrompt|0.6137|0.6697|0.7059|0.1567|
> > > >
> > > >
> > > > [19] Liu, Xiangyang, et al. "Late Prompt Tuning: A Late Prompt Could Be Better Than Many Prompts." Findings of the Association for Computational Linguistics: EMNLP 2022. 2022.
> > > >
> > > > BBT [20] is a gradient-free approach that is proposed for scenarios orthogonal to that in our paper. Specifically, it targets the scenarios where the pre-trained model is not available locally (i.e., deployed online) and its back-propagation operation is not available. For such cases, BBT is a way of gradient-free learning that sacrifices computation to avoid back-propagation, i.e., replacing back-propagation with tens of forward-propagation (please refer to the population size of [20]). In our paper, we consider training a local model that the back-propagation operation is available, for which case there is no need to sacrifice computation for gradient-free learning. This is also the commonly considered scenario in previous prompt tuning papers (e.g., [3-7]). This is why we exclude BBT for comparison.
> > > >
> > > > [20] Sun, Tianxiang, et al. "Black-box tuning for language-model-as-a-service." International Conference on Machine Learning. PMLR, 2022.
> > > >
> > > > <3. one of your contributions>
> > > > Thank you for the suggestion to improve our paper. Due to the limited space, it is difficult to include all the theoretical analysis in the main paper. We will summarize a proof sketch in the updated version, to better present our theoretical results.
> > > >
> > > > *** ***
> > > >
> > > > References mentioned in the earlier response:
> > > >
> > > > [1] An Information-theoretic Approach to Prompt Engineering Without Ground Truth Labels, ACL 2022
> > > >
> > > > [2] Liu, Pengfei, et al. "Pre-train, prompt, and predict: A systematic survey of prompting methods in natural language processing." ACM Computing Surveys 55.9 (2023): 1-35.
> > > >
> > > > [3] Li, X. L., & Liang, P. (2021, August). Prefix-Tuning: Optimizing Continuous Prompts for Generation. In Proceedings of the 59th Annual Meeting of the Association for Computational Linguistics and the 11th International Joint Conference on Natural Language Processing (Volume 1: Long Papers) (pp. 4582-4597).
> > > >
> > > > [4] Razdaibiedina, Anastasia, et al. "Progressive Prompts: Continual Learning for Language Models." The Eleventh International Conference on Learning Representations. 2022.
> > > >
> > > > [5] Wang, Zhen, et al. "Multitask Prompt Tuning Enables Parameter-Efficient Transfer Learning." The Eleventh International Conference on Learning Representations. 2022.
> > > >
> > > > [6] Razdaibiedina, Anastasia, et al. "Residual Prompt Tuning: Improving Prompt Tuning with Residual Reparameterization." arXiv e-prints (2023): arXiv-2305. (Findings of ACL2023)
> > > >
> > > > [7] Pang, Bo, et al. "SharPT: Shared Latent Space Prompt Tuning." Findings of the Association for Computational Linguistics: EACL 2023. 2023.
> > > >
> > > > [8] GPT-4 Technical Report, OpenAI.
> > > >
> > > > [9] Koubaa, Anis. "GPT-4 vs. GPT-3.5: A concise showdown." (2023).
> > > >
> > > > [10] Sun, Simeng, et al. "How does in-context learning help prompt tuning?." arXiv preprint arXiv:2302.11521 (2023).
> > > >
> > > > [11] Shi, Zhengxiang, and Aldo Lipani. "Don't Stop Pretraining? Make Prompt-based Fine-tuning Powerful Learner." arXiv preprint arXiv:2305.01711 (2023).

---

### Official Review · Reviewer_WbYf · 2023-07-06

**Soundness:** 3 good
**Presentation:** 3 good
**Contribution:** 2 fair
**Rating:** 6
**Confidence:** 4

**Summary:**

This paper introduces InfoPrompt: an information theoretic framework for soft prompt tuning. It does so by introducing two additional loss terms: the head loss (maximizing prompt similarity with the LM head) and the representation loss (maximizing prompt similarity with the encoder's last hidden state).
The authors show that InfoPrompt is better than WARP, IDPG, and Adapter baselines with Roberta Large as the base model, on sequence classification, RE, and NER datasets.
The paper includes analysis of loss landscapes and theoretical guarantees that their method can be optimized with gradient descent.

**Strengths:**

Strengths:
* The paper is clearly written, and is easy to follow.
* Infoprompt is simple to understand and implement.
* Infoprompt does better than the baselines it compares against: WARP, IDPG, Adapters and vanilla prompt tuning on Roberta Large.
* The paper illustrates interesting loss dynamics of their method in section 6.

**Weaknesses:**

Weaknesses:
* *Baselines*: I think there could be more baselines than WARPS and IDPG used here, like non-prompt-based parameter-efficient tuning methods besides Adapters (like LoRA or HyperFormer), or a more relevant one like HyperPrompt. I think this paper would be a valuable contribution even if it didn't completely beat the performance of those methods, but it would be nice to see how it compares to more diverse baselines (not necessarily the exact ones I described here, though they are certainly good choices).
* *Datasets*: The authors evaluated on most datasets in GLUE, it would be nice to have the full suite of GLUE evaluation. Having only a subset of GLUE feels like the datasets are cherry picked.
* *Tasks*: Some results on sequence generation tasks would have made this paper a lot stronger in my opinion, and it would not be significantly more expensive to train a T5 variant with this method on a summarization or translation task.
* *Minor typos* (did not affect review score):
    * Table 4 description
    * line 331
    * line 307

**Questions:**

Questions:
* Why is there no IDPG baseline in section 5.2?
* Prompt tuning (Lester et al) required a traditional causal LM training stage of 100k steps on T5 before prompt tuning for it to work well. The same issue might be hidering vanilla prompt tuning on the encoder-only models used here (Roberta variants). Given that the authors could train Roberta large, it seems that they have the compute required to train the released LM-adapted T5 variants in Lester et al [[1](https://huggingface.co/google/t5-base-lm-adapt), [2](https://github.com/google-research/text-to-text-transfer-transformer/blob/main/released_checkpoints.md##lm-adapted-t511lm100k)] for a better comparison to vanilla prompt tuning. Do the authors think that the prompt tuning baseline here (both InfoPrompt loss weights set to 0) might not be fairly compared to because of the misalignment of the base pretrained models used here?

**Limitations:**

Limitations:
* Evaluation is specific to Roberta large, and only sequence classification and RE/NER datasets. This does not diminish the intellectual value of the method, but is nonetheless a limitation that makes this method hard to confidently brand as better than vanilla prompt tuning.

---

> ### Author Rebuttal · Authors · 2023-08-10
>
> Thanks for your valuable comments!
>
> Weakness 1: ***Baselines***
>
> Response: Thanks for the suggestion. We further include the comparison between our approach and a new baseline LoRA, as below.
>
> | Full-data | CoLA | RTE | MRPC | SST2 | RE | NER | SemEval | Average |
> |---|---|---|---|---|---|---|---|---|
> | LoRA | 0.5880 | 0.6715 | 0.8235 | 0.9541 | 0.6636 | 0.8228 | 0.7214 | 0.7492 |
> | InfoPrompt | 0.6018 | 0.6968 | 0.8137 | 0.9599 | 0.7616 | 0.8962 | 0.7917 | 0.7888 |
>
> | 64 shots | CoLA | RTE | MRPC | SST2 | RE | NER | SemEval | Average |
> |---|---|---|---|---|---|---|---|---|
> | LoRA | 0.0991 | 0.5596 | 0.6985 | 0.5677 | 0.1232 | 0.1345 | 0.1711 | 0.3362 |
> | InfoPrompt | 0.1567 | 0.6137 | 0.7059 | 0.6697 | 0.2119 | 0.3331 | 0.2113 | 0.4146 |
>
> | 256 shots | CoLA | RTE | MRPC | SST2 | RE | NER | SemEval | Average |
> |---|---|---|---|---|---|---|---|---|
> | LoRA | 0.2854 | 0.5740 | 0.7206 | 0.8222 | 0.2291 | 0.1955 | 0.3817 | 0.4583 |
> | InfoPrompt | 0.1750 | 0.6580 | 0.7377 | 0.7305 | 0.2993 | 0.4739 | 0.4034 | 0.4968 |
>
> In our paper, we target improving the performance for a $\textbf{single task}$ via prompt tuning (similar to WARP[1], IDPG[3] and LoRA[4]).
> Therefore, we only include single task prompt tuning approaches as baselines.
> HyperFormer is originally designed for $\textbf{multi-task}$ learning, which is setting that is orthogonal to our paper.
> In our experiments, we follow the evaluation protocol in WARP[1], IDPG[3] and LoRA[4], and it is not feasible to directly compare HyperFormer in our designed evaluation protocol and experiment setting.
>
>
> Weakness 2: ***Dataset***
>
> Response: As explained in line 175, these 4 datasets chosen from the GLUE benchmark are relative small in size (line 175), which simulate a more challenging low-resource scenario that is compatible with prompt tuning.
>
>
> Weakness 3: ***Tasks***
>
> Response: In the paper, we follow previous works of prompt tuning, e.g., WARP [1], IDPG [3], etc., that mostly experiment with classification-based tasks. It would also be interesting to further consider sequence generation tasks. Thank you for the suggestion and we will keep this in mind in our future works.
>
>
> Weakness 4: ***Typos***
>
> Response: Thanks for pointing out our typos. We will fix them accordingly.
>
> Q1: ***Why is there no IDPG baseline in section 5.2?***
>
> A1: Similar to the evaluation in WARP [1], we only chose the most competitive baselines in full dataset experiments, in the further evaluations under the few-shot learning setting.
> Thanks for the suggestions. We further include the comparison between our approach and IDPG, in the few-shot learning setting.
>
> | 64 shots | CoLA | RTE | MRPC | SST2 | RE | NER | SemEval | Average |
> |---|---|---|---|---|---|---|---|---|
> | IDPG | 0.0902 | 0.5018 | 0.6593 | 0.5424 | 0.2596 | 0.3334 | 0.1984 | 0.3693 |
> | InfoPrompt | 0.1567 | 0.6137 | 0.7059 | 0.6697 | 0.2119 | 0.3331 | 0.2113 | 0.4146 |
>
> | 256 shots | CoLA | RTE | MRPC | SST2 | RE | NER | SemEval | Average |
> |---|---|---|---|---|---|---|---|---|
> | IDPG | 0.1513 | 0.5523 | 0.7010 | 0.8188 | 0.2503 | 0.4048 | 0.3577 | 0.4623 |
> | InfoPrompt | 0.1750 | 0.6580 | 0.7377 | 0.7305 | 0.2993 | 0.4739 | 0.4034 | 0.4968 |
>
> Q2:
> ***Do the authors think that the prompt tuning baseline here (both InfoPrompt loss weights set to 0) might not be fairly compared to because of the misalignment of the base pretrained models used here?***
>
> A2:
> Since we target classification-based tasks in the paper, we follow WRAP [1] and iPEC [2] that adopt an encoder-only LM (Roberta-Large). For fair comparison (following [1, 2]), all our baselines are implemented with the same base pretrained model (i.e, Roberta-Large).
>
> [1] Karen Hambardzumyan, Hrant Khachatrian, and Jonathan May. Warp: Word-level adversarial reprogramming. In Proceedings of the 59th Annual Meeting of the Association for Computational Linguistics and the 11th International Joint Conference on Natural Language Processing 385 (Volume 1: Long Papers), pages 4921–4933, 2021.
>
> [2] Schick, Timo, and Hinrich Schütze. "It’s Not Just Size That Matters: Small Language Models Are Also Few-Shot Learners." Proceedings of the 2021 Conference of the North American Chapter of the Association for Computational Linguistics: Human Language Technologies. 2021.
>
> [3] Wu, Zhuofeng, et al. "IDPG: An instance-dependent prompt generation method." arXiv preprint arXiv:2204.04497 (2022).
>
> [4] Hu, Edward J., et al. "Lora: Low-rank adaptation of large language models." arXiv preprint arXiv:2106.09685 (2021).

---

> > ### Comment · Reviewer_WbYf · 2023-08-14
> > **Response**
> >
> > Thank you for the rebuttal!
> >
> > I think weaknesses 1 and 4 have been addressed well, as well as question 1.
> >
> > I do however still think that this method could benefit from more diverse tasks for this paper to be a confident accept in my opinion. Showing that this method works well in generative tasks like language modeling or translation (or even span selection tasks like extractive question answering) would make it a strong paper in my opinion. Also, while I appreciate the focus on smaller GLUE datasets, there is still value in showing numbers on higher-resource tasks. Parameter efficient tuning and dataset size are orthogonal axes of study in my mind, so it would still be useful to see numbers on the full suite of GLUE tasks (or even a newer benchmark if possible, since GLUE is at a point of saturation).
> >
> > I think this paper is promising work and I encourage the authors to make it stronger. I will be keeping my review score the same.

---

> > > ### Author Response · Authors · 2023-08-21
> > >
> > > Thank you very much for the suggestion. We sincerely appreciate the time and effort you invested in the review. Though we follows the scope of previous prompt tuning works (e.g., [1][3][4]), it would certainly be interesting to further explore with more tasks. We will keep this in mind in our work.

---

### Official Review · Reviewer_kCoa · 2023-07-08

**Soundness:** 3 good
**Presentation:** 3 good
**Contribution:** 2 fair
**Rating:** 5
**Confidence:** 4

**Summary:**

The paper proposes a new formula that models soft prompt tuning as maximizing mutual information between prompts and other model parameters. In order to improve the initialization of the prompts and to learn sufficient task-relevant information from prompt tokens, this paper develops two novel mutual information based loss functions. The authors provide analysis on the convergence of the prompt tuning and the proposed methodology is evaluated on several benchmark datasets.

**Strengths:**

+ Simple but powerful idea: the authors propose a novel method for soft prompting.

+ Convincing and superior experimental results over baseline.

+ The paper is well-structured and easy to follow.

**Weaknesses:**

-Lack of ablation studies analysis: there are hyper-parameters that balance the weight of two loss function terms. However, I did not find any design choice and/or ablation study on this.

-In addition, more implementation details should be provided in the paper, which is essential for reproducing the methods and experiments in this work.

**Questions:**

Some detailed questions:

1. Figure 2(a) caption: confusion of definition: task loss info loss, representation loss? Line 257, the loss is mentioned as info loss but representation loss in the caption.

2. Line 73 experiments on 6 tasks and 3 datasets?

3. Could you explain the determination of β = 0.1 and γ = 0.05, the comparison of these two loss terms the number of negative samples and the importance of the number of prompt tokens

4. What are the advantages of the proposed method compared with baseline IDPG?
Why IDPG is utilized as a baseline method for full-dataset experiments while not utilized for few-shot dataset?

**Limitations:**

Yes.

---

> ### Author Rebuttal · Authors · 2023-08-10
>
> Thanks for your valuable comments!
>
> Weakness 1 & Q3: ***hyper-parameters ablation studies and the importance of the number of prompt tokens***
>
> Response:
> Similar to previous works [2, 3], we empirically choose the current hyper-parameters (without using the validation data). Specifically, based on our observation of the scales of the three losses, we choose the current hyper-parameters (without using validation data), so the scales of the three loss functions after weighting (head loss and representation loss, and task loss) are similar in the final loss function. This strategy is simply based on the inherent assumption that each loss should contribute equally to the problem [4].
>
> We improve on the previous soft prompt methods (e.g., WARP [1]) with our proposed novel information-theoretic objectives. Our choice of number of prompt tokens follows [1] and [5]. The submitted paper has already included results with different number of prompt tokens, detailed in Table 1 and 2. Similar to WARP [1], we observe that more prompt tokens generally lead to better performance.
>
>
> Weakness 2: ***more implementation details***
>
> Response:
> The implementation details of our proposed loss functions are as detailed in Section 3 and Section 4.2. Besides, we also include more details about the configurations (e.g., number of prompts, optimization details and networks design) in Section 4.2. Thanks for the suggestion. We will better organize the information for the reproducibility in the updated version.
>
>
> Q1: ***confusion of definition: task loss info loss, representation loss?***
>
> A1: Thanks for pointing this out. Line 257, the "info loss" should be the "representation loss".
>
>
> Q2: ***experiments on 6 tasks and 3 datasets?***
>
> A2: Sorry for the confusion. We experiment with 3 NLP tasks and 6 datasets.
>
> Q4: ***compared with baseline IDPG?***
>
> A4: Compared to IDPG, our approach has the advantages that we develop the information-theoretic method to better encode task-relevant information.
> Similar to the evaluation in WARP [1], we only chose the most competitive baselines (in full dataset experiments), in the further evaluations under the few-shot learning setting.
> Thanks for the suggestions. We further include the comparison between our approach and IDPG, in the few-shot learning setting.
>
> | 64 shots | CoLA | RTE | MRPC | SST2 | RE | NER | SemEval | Average |
> |---|---|---|---|---|---|---|---|---|
> | IDPG | 0.0902 | 0.5018 | 0.6593 | 0.5424 | 0.2596 | 0.3334 | 0.1984 | 0.3693 |
> | InfoPrompt | 0.1567 | 0.6137 | 0.7059 | 0.6697 | 0.2119 | 0.3331 | 0.2113 | 0.4146 |
>
> | 256 shots | CoLA | RTE | MRPC | SST2 | RE | NER | SemEval | Average |
> |---|---|---|---|---|---|---|---|---|
> | IDPG | 0.1513 | 0.5523 | 0.7010 | 0.8188 | 0.2503 | 0.4048 | 0.3577 | 0.4623 |
> | InfoPrompt | 0.1750 | 0.6580 | 0.7377 | 0.7305 | 0.2993 | 0.4739 | 0.4034 | 0.4968 |
>
> We can observe that InfoPrompt generally yields better performance than IDPG due to its advantage to better encode task-relevant information.
>
> [1] Karen Hambardzumyan, Hrant Khachatrian, and Jonathan May. Warp: Word-level adversarial reprogramming. ACL, 2021.
>
> [2] Jia, Zhiwei, and Hao Su. "Information-theoretic local minima characterization and regularization." International Conference on Machine Learning. PMLR, 2020.
>
> [3] Shi, Yufeng et al. “Information-Theoretic Hashing for Zero-Shot Cross-Modal Retrieval.” ArXiv abs/2209.12491 (2022): n. pag.
>
> [4] Groenendijk, Rick, et al. "Multi-loss weighting with coefficient of variations." Proceedings of the IEEE/CVF winter conference on applications of computer vision. 2021.
>
> [5] Zhou, Yuhang, Suraj Maharjan, and Beiye Liu. "Scalable Prompt Generation for Semi-supervised Learning with Language Models." EACL 2023.

---

### Decision · Program_Chairs · 2023-09-21

**Decision:**

Accept (poster)

**Comment:**

The paper presents a new method for soft prompt tuning in NLU tasks, using mutual information-based loss functions. It shows superior performance over existing baselines but lacks thorough ablation studies and diverse comparisons. The work is incremental and could benefit from broader evaluations and more detailed implementation information.

The reviewers are generally all satisfied with the authors' responses. The reviewer who insists on borderline reject agreed to accept the paper as poster.

AC reads the paper (roughly), the review, and the discussion, and recommends accept (poster).